# NVS-Solver: Video Diffusion Model as Zero-Shot Novel View Synthesizer

**Meng You[1†], Zhiyu Zhu[1†*], Hui Liu[2] & Junhui Hou[1]**
[1]City University of Hong Kong, [2]Saint Francis University
{mengyou2, zhiyuzhu2-c}@my.cityu.edu.hk
h2liu@sfu.edu.hk  jh.hou@cityu.edu.hk

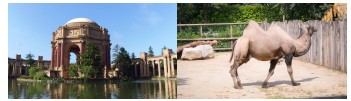

|  |  |  |
|:---:|:---:|:---:|
| (a) | (b) | (c) |

Figure 1: Visual demonstrations of the proposed algorithm with input of (**a**) single-view, (**b**) monocular video, and (**c**) multi-view (2 views). The middle row refers to the algorithm's inputs for each scenario. Please use *Adobe Acrobat* to display these videos.

## ABSTRACT

By harnessing the potent generative capabilities of pre-trained large video diffusion models, we propose a new novel view synthesis paradigm that operates *without* the need for training. The proposed method adaptively modulates the diffusion sampling process with the given views to enable the creation of visually pleasing results from single or multiple views of static scenes or monocular videos of dynamic scenes. Specifically, built upon our theoretical modeling, we iteratively modulate the score function with the given scene priors represented with warped input views to control the video diffusion process. Moreover, by theoretically exploring the boundary of the estimation error, we achieve the modulation in an adaptive fashion according to the view pose and the number of diffusion steps. Extensive evaluations on both static and dynamic scenes substantiate the significant superiority of our method over state-of-the-art methods both quantitatively and qualitatively. The source code can be found on https://github.com/ZHU-Zhiyu/NVS_Solver.

## 1 INTRODUCTION

In the realm of computer vision and graphics, novel view synthesis (NVS) from limited visual data remains a formidable challenge with profound implications across various domains, from

---

[†]Equal Contribution.

[*]Corresponding Author. This project was supported in part by the NSFC Excellent Young Scientists Fund 62422118, in part by the Hong Kong RGC under Grants 11219422 and 11219324, and in part by Hong Kong UGC under grants UGC/FDS11/E02/22 and UGC/FDS11/E03/24.

entertainment (Tewari et al., 2020; Jiang et al., 2024) to autonomous navigation (Adamkiewicz et al., 2022; Kwon et al., 2023) and beyond (Avidan & Shashua, 1997; Zhou et al., 2016; Riegler & Koltun, 2020; Mildenhall et al., 2021; Kerbl et al., 2023). However, addressing this challenge demands a sophisticated method capable of extracting meaningful information from sparse visual inputs and synthesizing coherent representations of unseen viewpoints (Zou et al., 2024; Zhang et al., 2021). In this context, the emerging field of deep learning has witnessed remarkable strides, particularly with the advent of advanced generative models (Voleti et al., 2024; Chen et al., 2023; Gu et al., 2023).

Recently, diffusion models (Ho et al., 2020; Song et al., 2020; 2021b) have garnered significant attention due to their exceptional ability to synthesize visual data. A prominent area of focus within this domain is video diffusion models (Blattmann et al., 2023; Khachatryan et al., 2023; Ho et al., 2022; Ni et al., 2023; Karras et al., 2023; Khachatryan et al., 2023; Wu et al., 2023), which have gained popularity for their remarkable video generation capabilities. In this paper, we explore the problem of NVS from single or multiple views of static scenes or monocular videos of dynamic scenes, leveraging the pre-trained large video diffusion model *without additional training*. Specifically, we theoretically formulate the NVS-oriented diffusion process as guided sampling, in which the intermediate diffusion results are modulated with the scene information from the given views. Moreover, we empirically and theoretically investigate the potential distribution of the error map to achieve *adaptive* modulation in the reverse diffusion process, with a reduced estimation error boundary.

In summary, the main contributions of this paper lie in:

- we propose a new *training-free* novel view synthesis paradigm by leveraging pre-trained video diffusion models;
- we theoretically formulate the process of *adaptively* utilizing the given scene information to control the video diffusion process; and
- we demonstrate the remarkable performance of our paradigm under various scenarios.

## 2 RELATED WORK

**Diffusion Model** indicates a kind of deep generative model (Sohl-Dickstein et al., 2015), which is inspired by non-equilibrium statistical physics by iteratively appending noise into image data and then reversely removing noise and transferring to noise-free data distribution. (Ho et al., 2020) proposed a variance preserving (VP) diffusion denoising probabilistic model via progressively removing noise. (Song et al., 2021b; Song & Ermon, 2019; 2020; Song et al., 2021a) proposed a score-based image generation model by iteratively calculating the derivative of data distribution and utilizing the stochastic differential equation (SDE) (Anderson, 1982) or ordinary differential equation (ODE)-based solvers (Maoutsa et al., 2020; Song et al., 2021b) to reverse the noise-adding process and derive the clean image distribution. Inspired by the success of image diffusion models, many works attempted to build video diffusion model to directly achieve video generation prompted by text (Khachatryan et al., 2023; Wu et al., 2023) or a single image (Blattmann et al., 2023; Karras et al., 2023).

**Diffusion sampling algorithm** aims to speed up or control the diffusion process via regularizing or re-directing the reverse trajectory of diffusion models (Lu et al., 2022a;b; Zheng et al., 2024; Zhang & Chen, 2022; Wang et al., 2023b; Cao et al., 2024; Chung et al., 2023; 2022a;b). (Song et al., 2020) proposed denoising diffusion implicit models (DDIM) to accelerate the diffusion sampling by jumping to the clean image-space at each step. (Dhariwal & Nichol, 2021) utilized the classifier to guide the sampling process of diffusion model for controlling the results' categories. (Lu et al., 2022a;b; Zheng et al., 2024) proposed a series of fast ODE diffusion solvers given an analytic solution of ODE by its semi-linear nature. Moreover, the integration of non-linear network-related parts was further approximated by its Taylor series. (Zhang & Chen, 2022) explored the huge variance of distribution shift and then decoupled an exponential variance component from the score estimation model, thus reducing the discretization error. (Chung et al., 2023) proposed to regularize the intermediate derivative from the reconstruction process to achieve image restoration. (Wang et al., 2023b) decoupled the image restoration into range-null spaces and focused on the reconstruction of null space, which contains the degraded information.

**Novel view synthesis (NVS)** targets at generating images of a scene from viewpoints not presented in the original data and has been a subject of extensive research in computer vision and graphics (Park et al., 2017; Choi et al., 2019; Tretschk et al., 2021; You et al., 2023; Avidan & Shashua, 1997;

Riegler & Koltun, 2021; You & Hou, 2024). Early methods in this domain often relied on geometric approaches, such as multi-view stereo reconstruction (Seitz et al., 2006; Jin et al., 2005) and structure-from-motion (Schonberger & Frahm, 2016; Özyeşil et al., 2017) techniques, to synthesize novel viewpoints from multiple images captured from different angles. The advent of deep learning has revolutionized the field of NVS, enabling the synthesis of realistic images from pre-trained feature volumes. (Rombach et al., 2021; Ren & Wang, 2022) employed autoregressive Transformer to synthesize 3D scene from single image. Neural Radiance Fields (NeRF) (Mildenhall et al., 2021; Pumarola et al., 2021; Barron et al., 2021; Martin-Brualla et al., 2021; Kosiorek et al., 2021) enabled stunningly detailed renders from 2D images through volumetric rendering. Additionally, differentiable rendering has allowed gradients of rendering outputs with respect to scene parameters, facilitating direct optimization of scene geometries, lighting, and materials. Recently, 3D Gaussian Splatting (Kerbl et al., 2023; Liu et al., 2024) presented an explicit representation of the scene.

While these methods have shown promising results, they often suffer from limitations such as dependence on dense scene geometry, challenges in handling complex scene dynamics, and being hard to generalize (Kerbl et al., 2023). Moreover, they require significant computational resources and suffer from artifacts such as disocclusions and view-dependent effects. More recently, (Charatan et al., 2024; Chen et al., 2025) introduced the use of feed-forward 3D Gaussians for novel view interpolation. (Wu et al., 2024b; Watson et al., 2023; Yu et al., 2023; Cai et al., 2023; Tseng et al., 2023; Chan et al., 2023; Sargent et al., 2024) explored the integration of 3D reconstruction techniques with diffusion models.

In light of these challenges, our work builds upon the strengths of recent advancements in deep learning-based NVS, with a focus on leveraging the robust zero-shot view synthesis capabilities of the video diffusion model. By harnessing latent representations derived from sparse or single-view inputs, our approach aims to overcome the limitations of existing methods and produce high-quality novel views with improved realism.

## 3  PRELIMINARY OF DIFFUSION MODELS

We briefly introduce some preliminary knowledge of diffusion models (Sohl-Dickstein et al., 2015), which facilitates our subsequent analyses. We also refer readers to (Song & Ermon, 2019; Song et al., 2021b) for more details. Generally, the forward SDE process of the latent diffusion model can be formulated as

$$dx = f(t)x dt + g(t)dw, \tag{1}$$

where $x$ is the noised latent state, $t$ for the timestamp of diffusion, and the two scalar functions $f(t)$ and $g(t)$ output drift and diffusion coefficients, indicating the variation of data and noise components during the diffusion process, respectively. Naturally, we have the following ODE solution:

$$dx = \left[ f(t)x - \frac{1}{2}g^2(x)\nabla_x \log(q_t(x)) \right] dt. \tag{2}$$

Through training a score model $S_\theta(x, t)$ parameterized with $\theta$ to approximate the $\nabla_x \log[q(x)]$ as

$$\mathcal{L} = \gamma(t)\|S_\theta(x, t) - \nabla_x \log[q_t(x)]\|_2^2, \tag{3}$$

with $\gamma(t) > 0$, we can calculate the clean image $x_0$ via utilizing the score function $S_\theta(x, t)$ to calculate the data distribution gradient as

$$dx = \left[ f(t)x - \frac{1}{2}g^2(x)S_\theta(x, t) \right] dt. \tag{4}$$

Moreover, since the noise of the diffusion process is usually parameterized by i.i.d. Gaussian noises $\mathcal{N}(\epsilon_t; 0, \sigma(t)\mathbf{I})$ with a variance of $\sigma(t)$, the above diffusion process can be calculated as

$$dx = \left[ f(t)x - \frac{1}{2}g^2(x)\frac{\mu_t - x}{\sigma^2(t)} \right] dt, \tag{5}$$

where $\mu_t = \mathcal{X}_\theta(x_t, t)$ is the estimated clean image from the noised image $x_t$ at step $t$ by the denosing U-Net $\mathcal{X}_\theta(\cdot, \cdot)$[1]. Finally, considering the special case of variance exploding (VE) diffusion process (Song et al., 2021b) of the stable video diffusion (SVD) (Blattmann et al., 2023), Eq. (5) can be further simplified as

$$dx = \frac{x - \mu_t}{\sigma(t)}d\sigma(t). \tag{6}$$

---

[1]For simplicity, here we combine some parameterizations stemmed from EDM (Karras et al., 2022) into $\mathcal{X}_\theta(\cdot)$.

# 4 PROPOSED METHOD

Motivated by the powerful generative capability with realistic and consistent frames by large video diffusion models, we aim to adapt the pre-trained video diffusion model to the task of NVS ***without any additional training***, leading to a score modulation-based approach. Generally, our method warps the input image set to the target view and leverages these warped images as prior information to guide the reverse sampling of a video diffusion process. This not only ensures the accurate generation of the warped regions but also facilitates the meaningful synthesis of unknown or occluded areas, resulting in high-quality and visually consistent novel views with enhanced fidelity and coherence. Specifically, we first theoretically reformulate NVS-oriented reverse diffusion sampling by modulating the score function with the given views (Sec. 4.1). Based on theoretically exploring the boundary of the diffusion estimation and depth-based warping errors, we propose to adaptively modulate the prior of the given view into the diffusion process to reduce the potential boundary of the estimation error (Sec. 4.2).

In the following, we take the single view-based NVS to illustrate our method, as summarized in Algorithm 1, which involves from a given view $\mathbf{X}_{0,\mathbf{p}_0}$ with pose $\mathbf{p}_0$, reconstructing $N-1$ novel views at target poses $\{\mathbf{p}_1, \cdots, \mathbf{p}_i, \cdots, \mathbf{p}_{N-1}\}$, denoted as $\mathbf{X}_{0,\mathbf{p}_i}$, . Note that our method is suitable for scenarios involving NVS from multiple views, as well as from monocular videos, as explained in Sec. 4.3. Also, we consider the pre-trained image-to-video diffusion model SVD (Blattmann et al., 2023).

**Notations**. Let $\mathbf{X}_t \in \mathbb{R}^{H \times W \times N}$ be the spatial-temporal latent of a set of $N$ images/views at the $t$-th diffusion step, $\mathbf{p}_i \in \mathbb{R}^5$ the pose of the $i$-th image $(0 \leq i \leq N-1)$, $\mathcal{X}_{\boldsymbol{\theta}}(\mathbf{X}_t, t)$ a U-Net denoising $\mathbf{X}_t$ to estimate the clean image set $\boldsymbol{\mu}_t \in \mathbb{R}^{H \times W \times N}$, $\mathbf{X}_{t,\mathbf{p}_i} \in \mathbb{R}^{H \times W}$ a typical latent at step $t$, time $i$, and spatial pose $\mathbf{p}_i$, and $\boldsymbol{\mu}_{t,\mathbf{p}_i} = \mathcal{X}_{\boldsymbol{\theta}}^{\mathbf{P}_i}(\mathbf{X}_t, t)$, indicating a intermediate estimation of clean image $\mathbf{X}_{0,\mathbf{p}_i}$.

## 4.1 SCENE PRIOR MODULATED REVERSE DIFFUSION SAMPLING

Based on the formulation of the image reverse diffusion sampling process in Eq. (6), we have

$$d\mathbf{X}_{t,\mathbf{p}_i} = \left[ \frac{\mathbf{X}_{t,\mathbf{p}_i} - \mathcal{X}_{\boldsymbol{\theta}}^{\mathbf{P}_i}(\mathbf{X}_t, t)}{\sigma(t)} \right] d\sigma(t). \tag{7}$$

According to the intensity function (McMillan & Bishop, 1995), we can formulate the relationship between $\mathbf{X}_{0,\mathbf{p}_0}$ and $\mathbf{X}_{0,\mathbf{p}_i}$ through Taylor expansion:

$$\mathbf{X}_{0,\mathbf{p}_i} = \mathcal{I}(\mathbf{p}_0) + \frac{d\mathcal{I}(\mathbf{p})}{d\mathbf{p}}\Delta\mathbf{p} + \mathcal{O}^2(\Delta\mathbf{p}), \tag{8}$$

where $\mathcal{I}(\cdot)$ is the intensity function, $\Delta\mathbf{p} := \mathbf{p}_i - \mathbf{p}_0$ is the pose variation, and $\mathcal{O}^2(\Delta\mathbf{p})$ is the high order Taylor expansions. Based on the depth-driven image-warping operation, we further have

$$\mathcal{I}(\mathbf{p}_0) = \mathcal{W}\left(\mathbf{X}_{0,\mathbf{p}_0}, \mathbf{u}_i\right), \; \mathbf{u}_i = \mathbf{K}\mathbf{D}\Delta\mathbf{p}\mathbf{K}^{-1}\mathbf{u}_0, \tag{9}$$

where $\mathcal{W}(\cdot, \cdot)$ denotes the image warping function; $\mathbf{u}_0$ and $\mathbf{u}_i$ are the pixel locations of the views at poses $\mathbf{p}_0$ and $\mathbf{p}_i$, respectively; $\mathbf{D}$ is the depth map; $\mathbf{K}$ is the camera intrinsic matrix. Moreover, since the ground-truth depth $\mathbf{D}$ is usually not available in practice, we can also use the estimated depth map $\widehat{\mathbf{D}}$ by an off-the-shelf depth estimation method to calculate the estimated pixel location, i.e., $\widehat{\mathbf{u}}_i = \mathbf{K}\widehat{\mathbf{D}}\Delta\mathbf{p}\mathbf{K}^{-1}\mathbf{u}_0$. By substituting Eq. (9) with an estimated depth map to Eq. (8), we then have

$$\mathbf{X}_{0,\mathbf{p}_i} = \underbrace{\mathcal{W}\left(\mathbf{X}_{0,\mathbf{p}_0}, \widehat{\mathbf{u}}_i\right)}_{\widehat{\mathbf{X}}_{0,\mathbf{p}_i}} + \underbrace{\frac{\partial\mathcal{W}\left(\mathbf{X}_{0,\mathbf{p}_0}, \widehat{\mathbf{u}}_i\right)}{\partial\mathbf{u}}\Delta\mathbf{u} + \frac{d\mathcal{I}(\mathbf{p})}{d\mathbf{p}}\Delta\mathbf{p} + \mathcal{O}^2(\Delta\mathbf{p})}_{\mathcal{E}_T}, \; \Delta\mathbf{u} = \mathbf{K}\Delta\mathbf{D}\Delta\mathbf{p}\mathbf{K}^{-1}\mathbf{u}_0,$$

$$\tag{10}$$

where $\Delta\mathbf{D} := \mathbf{D} - \widehat{\mathbf{D}}$, $\widehat{\mathbf{X}}_{0,\mathbf{p}_i}$ refers to the warped view from $\mathbf{p}_0$ to pose $\mathbf{p}_i$ through $\widehat{\mathbf{D}}$, and $\mathcal{E}_T$ is a residual term that contains high-order Taylor expansion series and warping error.

Based on the above analysis that the rendered novel view at pose $\mathbf{p}_i$ is highly correlated with $\widehat{\mathbf{X}}_{0,\mathbf{p}_i}$ and $\boldsymbol{\mu}_{t,\mathbf{p}_i}$, an effective score function for NVS should take advantage of them. Then, we formulate the score function of NVS-oriented reverse diffusion sampling, which is modulated with the given scene information as

$$\widetilde{\boldsymbol{\mu}}_{t,\mathbf{p}_i} = \arg\min_{\boldsymbol{\mu}} \|\boldsymbol{\mu} - \boldsymbol{\mu}_{t,\mathbf{p}_i}\|_2^2 + \lambda(t, \mathbf{p}_i)\|\boldsymbol{\mu} - \widehat{\mathbf{X}}_{0,\mathbf{p}_i}\|_2^2, \tag{11}$$

---

**Algorithm 1** Zero-shot NVS from Single Images

---

1: **Input:** given view $\{\mathbf{X}_{\mathbf{p}_0}\}$, estimated depth map $\widehat{\mathbf{D}}_{\mathbf{p}_0}$, target view poses $\{\mathbf{p}_0, \cdots, \mathbf{p}_{N-1}\}$, and diffusion U-Net $\mathcal{X}_{\boldsymbol{\theta}}(\cdot)$.
2: Derive the warping component $\{\widehat{\mathbf{X}}_{0,\mathbf{p}_0}, \cdots, \widehat{\mathbf{X}}_{0,\mathbf{p}_i}, \cdots, \widehat{\mathbf{X}}_{0,\mathbf{p}_{N-1}}\}$, using the given views, corresponding poses and depth maps by Eq. (10).   ▷ Preparing diffusion guidance images.
3: Initialize $\mathbf{X}_T \sim \mathcal{N}(\mathbf{0}, \sigma_t \mathbf{I})$
4: **For** $t = T, ..., 1$ **do**   ▷ Video diffusion reverse sampling steps.
5:    Diffusion network forward $\boldsymbol{\mu}_t = \mathcal{X}_{\boldsymbol{\theta}}(\mathbf{X}_t, t)$.
6:    **For** $i = 0, \cdots, N-1$ **do**
7:       Calculate weights $\widehat{\lambda}(t, \mathbf{p})$ via Eq. (17) and $\widetilde{\boldsymbol{\mu}}_{t,\mathbf{p}}$ via Eq. (12).
8:       **If** Directly guided sampling **then**
9:          Reverse $\mathbf{X}_{t,\mathbf{p}}$ to $\mathbf{X}_{t-1,\mathbf{p}}$ via Eq. (13).
10:       **End If**
11:    **End for**
12:    **If** Posterior sampling **then**
13:       Derive the optimized $\mathbf{X}_t'$ via Eq. (14) and apply a standard reverse step as Eq. (7) to get the $\mathbf{X}_{t-1}$.
14:    **End If**
15: **End for**
16: **Return** Reconstructed sequence $\mathbf{X}_0$.

---

where $\lambda(t, \mathbf{p}_i) > 0$ balances the two terms. It is obvious the closed-form solution of Eq. (11) is

$$\widetilde{\boldsymbol{\mu}}_{t,\mathbf{p}_i} = \frac{1}{1 + \lambda(t, \mathbf{p}_i)} \boldsymbol{\mu}_{t,\mathbf{p}_i} + \frac{\lambda(t, \mathbf{p}_i)}{1 + \lambda(t, \mathbf{p}_i)} \widehat{\mathbf{X}}_{0,\mathbf{p}_i}. \tag{12}$$

With the optimized clean image expectation term $\widetilde{\boldsymbol{\mu}}_{t,\mathbf{p}_i}$, we further utilize two ways to guide the reverse sampling of the pre-trained SVD: (**1**) *Directly Guided Sampling*, which is fast with limited quality; and (**2**) *Posterior Sampling*, which is slow but more effective. Specifically, the former directly replaces the $\boldsymbol{\mu}_{t,\mathbf{p}}$ in 7 with $\widetilde{\boldsymbol{\mu}}_{t,\mathbf{p}_i}$ as

$$d\mathbf{X}_{t,\mathbf{p}_i} = \left[\frac{\mathbf{X}_{t,\mathbf{p}_i} - \widetilde{\boldsymbol{\mu}}_{t,\mathbf{p}_i}}{\sigma(t)}\right] d\sigma(t). \tag{13}$$

While the latter embeds the knowledge of $\widetilde{\boldsymbol{\mu}}_t$ into $\mathbf{X}_t$ via the backward gradient as

$$\mathbf{X}_t' = \mathbf{X}_t - \kappa \nabla_{\mathbf{X}_t} \|\boldsymbol{\mu}_t - \widetilde{\boldsymbol{\mu}}_t\|_2, \tag{14}$$

where $\kappa > 0$ controls the updating rate, which is empirically set to $2e^{-2}\sqrt{\sigma(t)}$; $\nabla_{\mathbf{X}_t} \|\boldsymbol{\mu}_t - \widetilde{\boldsymbol{\mu}}_t\|_2$ computes the back-propagated gradient on $\mathbf{X}_t$; $\mathbf{X}_t'$ stands for the optimized noised latent $\mathbf{X}_t$. Here, we also normalize the gradient $\nabla_{\mathbf{X}_t} \|\boldsymbol{\mu}_t - \widetilde{\boldsymbol{\mu}}_t\|_2$ to stabilize the updating process.

## 4.2 ADAPTIVE DETERMINATION OF $\lambda(t, \mathbf{p}_i)$

Although we have reformulated the NVS-oriented diffusion process in the preceding section, the value of $\lambda(t, \mathbf{p}_i)$ in Eq. (11) and Eq. (12), which is crucial to the quality of synthesized views, has to be appropriately determined. Here, we theoretically explore the formulation of $\lambda(t, \mathbf{p}_i)$ via analyzing the upper boundary of the estimation error of $\widetilde{\boldsymbol{\mu}}_{t,\mathbf{p}_i}$.

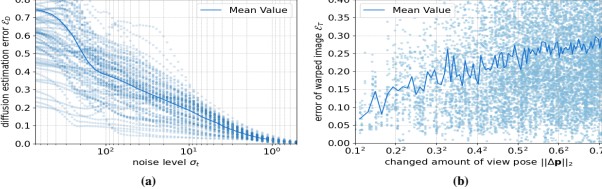

Figure 2: Experimental observations of the relationship (**a**) between the diffusion estimation error $\mathcal{E}_D$ and the noise level $\sigma_t$ and (**b**) between the error of warped image $\mathcal{E}_T$ and the changed amount of view pose $\|\Delta\mathbf{p}\|_2$.

Let $\widetilde{\boldsymbol{\mu}}_{t,\mathbf{p}_i}^*$ be the ideal value for the score function estimation in the NVS-oriented diffusion process, i.e., the ground-truth view at $p_i$. Based on the formulation of $\widetilde{\boldsymbol{\mu}}_{t,\mathbf{p}_i}$ in

Eq. (12) and the triangle inequality, we have

$$\|\widetilde{\boldsymbol{\mu}}_{t,\mathbf{p}_i} - \widetilde{\boldsymbol{\mu}}^*_{t,\mathbf{p}_i}\|_2 \leq \frac{1}{1+\lambda(t,\mathbf{p}_i)}\|\boldsymbol{\mu}_{t,\mathbf{p}_i} - \widetilde{\boldsymbol{\mu}}^*_{t,\mathbf{p}_i}\|_2 + \frac{\lambda(t,\mathbf{p}_i)}{1+\lambda(t,\mathbf{p}_i)}\|\widehat{\mathbf{X}}_{0,\mathbf{p}_i} - \widetilde{\boldsymbol{\mu}}^*_{t,\mathbf{p}_i}\|_2. \quad (15)$$

Thus, we propose to minimize the estimation error upper bound to obtain an appropriate and adaptive $\lambda(t, \mathbf{p}_i)$, i.e.,

$$\widehat{\lambda}(t,\mathbf{p}_i) = \underset{\lambda(t,\mathbf{p}_i)}{\arg\min} \frac{1}{1+\lambda(t,\mathbf{p}_i)}\mathbb{E}_{\mathbf{X}_t \sim \mathcal{P}(\mathbf{X}_t)}(\mathcal{E}_D) + \frac{\lambda(t,\mathbf{p}_i)}{1+\lambda(t,\mathbf{p}_i)}\mathbb{E}_{\mathbf{X} \sim \mathcal{P}(\mathbf{X})}(\mathcal{E}_P) + v_1|\log(\lambda(t,\mathbf{p}_i))|, \quad (16)$$

where $v_1 > 0$ and the last regularization term prevents the weights from being overfitting on the empirically estimated errors. In the following, we will provide the explicit formulations of the two error terms $\mathcal{E}_D = \|\boldsymbol{\mu}_{t,\mathbf{p}_i} - \widetilde{\boldsymbol{\mu}}^*_{t,\mathbf{p}_i}\|_2$ and $\mathcal{E}_P = \|\widehat{\mathbf{X}}_{0,\mathbf{p}_i} - \widetilde{\boldsymbol{\mu}}^*_{t,\mathbf{p}_i}\|_2$ [2], based on theoretical analyses and experimental observations.

**Diffusion Estimation Error $\mathcal{E}_D$.** This error is caused due to the fact that the diffusion model cannot perfectly estimate $\widetilde{\boldsymbol{\mu}}^*_{t,\mathbf{p}_i}$ from the given noised latent $\mathbf{X}_t$, Recent works (Zhang & Chen, 2022; Zheng et al., 2024) indicate that the derivative $S_\theta(\mathbf{X}_t, t)$ of SDE-based diffusion models varies intensely when $\sigma(t)$ is huge. Accordingly, large values of $\sigma$ would potentially lead to large errors. Moreover, we experimentally investigated the correlation of $\|\mathcal{E}_D\|_2$ with $\sigma(t)$. As demonstrated in in Fig. 2 (a), $\|\mathcal{E}_D\|_2$ gradually decreases with the diffusion reverse process (decreasing of $\sigma(t)$). Thus, we empirically formulate $\|\mathcal{E}_D\|_2 = v_2\sigma(t)$, where $0 < v_2$.

**Intensity Truncation Error $\mathcal{E}_P$.** This error is mainly induced by the truncation of the Taylor series (here $\widetilde{\boldsymbol{\mu}}^*_{t,\mathbf{p}_i}$ is the same with $\mathbf{X}$), as shown in Eq. (10). If omitting the high-order terms, we have $\mathcal{E}_P \approx \frac{\partial \mathcal{W}(\mathbf{X}_{0,\mathbf{p}_0},\widehat{\mathbf{u}}_i)}{\partial \mathbf{u}}\Delta\mathbf{u} + \frac{d\mathcal{I}(\mathbf{p})}{d\mathbf{p}}\Delta\mathbf{p}$. According to the definition of $\Delta\mathbf{u}$ in Eq. (10), we also have $\|\mathcal{E}_p\|_2 \leq v\|\Delta\mathbf{p}\|_2$. Together with the experimental observation of the relationship between $\|\mathcal{E}_p\|_2$ and $\|\Delta\mathbf{p}\|_2$ in Fig. 2 (b), we empirically formulate $\|\mathcal{E}_p\|_2 = v_3\|\Delta\mathbf{p}\|_2$.

Finally, with the explicit formulations of the two error terms, we can rewrite Eq. (16) as

$$\widehat{\lambda}(t,\mathbf{p}_i) = \underset{\lambda(t,\mathbf{p}_i)}{\arg\min} \frac{v_2\sigma(t)}{1+\lambda(t,\mathbf{p}_i)} + \frac{\lambda(t,\mathbf{p}_i)v_3\|\Delta\mathbf{p}\|_2}{1+\lambda(t,\mathbf{p}_i)} + v_1|\log(\lambda(t,\mathbf{p}_i))|, \quad (17)$$

whose closed-form solution is (we have visualized $\widehat{\lambda}$ in Appendix D)

$$\widehat{\lambda}(t,\mathbf{p}_i) = \frac{-(2v_1+Q) + \sqrt{Q^2 + 4v_1 Q}}{2v_1}, \quad Q = v_3\|\Delta\mathbf{p}\|_2 - v_2\sigma(t). \quad (18)$$

**Claim of Novelty.** Our method presents a theoretical analysis that links NVS with diffusion processes. Examining the relationships between different views and assessing their potential error distributions enable adaptive modulation of the diffusion score function, effectively minimizing errors from both the warping operator and the diffusion process. This makes our method uniquely suited to addressing NVS challenges. *In contrast*, traditional guided sampling algorithms typically rely on image degradation models, such as blurring and noise, to guide the diffusion process, setting our approach apart. Moreover, we have theoretically discussed that the proposed method regularizes the diffusion trajectories towards a more accurate direction in *Appendix A.1*. We have also proved the proposed method (DGS) can safely regularize the samples on the data manifolds in *Appendix A.2*. We also experimentally compare with inpainting-based methods in *Appendix G*.

## 4.3 NVS FROM MULTIVIEWS AND MONOCULAR VIDEOS

We have illustrated the single view-based NVS via the proposed method in Algorithm 1, which can be further modified for NVS from multiple views or monocular videos. Specifically, given multiple views of a static scene and target poses, we first estimate the depth map of each of the given views and derive the warped view at target poses by warping the nearest given views to each target pose. For a monocular video, we warp each frame of the video sequence to the corresponding target pose with the same timestamp (*See the Appendix E for the detailed warp strategy pipeline*). Then, we sample the novel view via the proposed method as Lines 3-16 of Algorithm 1.

---

[2]There is also inherent discretization error as investigated by (Lu et al., 2022a;b; Zhang & Chen, 2022). However, we could reduce such error terms by enlarging the number of reversing steps.

Table 1: Quantitative comparison of different methods on view synthesis, where we measure the FID and the error of view pose. We name our method with Ours (DGS) or (Post) for utilizing directly guided sampling in Eq. (13) or posterior sampling in Eq. (14), respectively. "*" indicates **incomplete** evaluation, where the corresponding metric cannot output a meaningful value on some sequences by the method due to the failure of pose estimation. "– –" denotes that the method cannot work on the condition or the metrics cannot be calculated. *For all metrics, the lower, the better.*

| Methods | Overfitting | Single view | | | | Multi-view | | | |
|---|---|---|---|---|---|---|---|---|---|
| | | FID | ATE | RPE-T | RPE-R | FID | ATE | RPE-T | RPE-R |
| Sparse Gaussian (Xiong et al., 2023) | ✓ | 369.19 | – – | – – | – – | 324.60 | – – | – – | – – |
| Sparse Nerf (Wang et al., 2023a) | ✓ | – – | – – | – – | – – | 180.26 | 6.129 | 1.711 | 1.804 |
| Text2Nerf (Zhang et al., 2024) | ✓ | 187.05 | 2.223* | 0.718* | 0.107* | – – | – – | – – | – – |
| Photoconsistent-NVS (Yu et al., 2023) | ✗ | 193.87 | 7.64 | 1.19 | 1.45 | – – | – – | – – | – – |
| 3D-aware (Xiang et al., 2023) | ✗ | 217.19 | 2.836 | 1.258 | 1.662 | 211.25 | 2.159* | 5.679* | 2.119 * |
| MotionCtrl (Wang et al., 2024) | ✗ | 179.24 | 3.851 | 0.705 | 0.835 | 154.27 | 37.68 | 19.61 | 1.646 |
| **Ours (DGS)** | ✗ | 166.50 | 4.533 | 0.810 | 0.742 | 124.31 | 22.00 | 17.34 | 1.338 |
| **Ours (Post)** | ✗ | 165.12 | 0.767 | 0.156 | 0.170 | 126.44 | 4.052 | 2.030 | 0.330 |

## 5 EXPERIMENT

**Datasets.** For *single view-based NVS*, we employed a total of nine scenes, with six scenes from the *Tanks and Temples* dataset (Knapitsch et al., 2017), containing both outdoor and indoor environments. The other three additional scenes are randomly chosen from the Internet. For *multiview-based NVS*, we used three scenes from the *Tanks and Temples* dataset (Knapitsch et al., 2017), including both outdoor and indoor settings, as well as six scenes from the *DTU* dataset (Jensen et al., 2014), which feature indoor objects. For each scene, we selected two images as input to perform view interpolation. For *monocular video-based NVS*, we downloaded nine videos from YouTube, each comprising 25 frames and capturing complex scenes in both urban and natural settings.

**Implementation Details.** We conducted all the experiments with PyTorch using a single NVIDIA GeForce RTX A6000 GPU-48G. We adopted the point-based warping (Somraj, 2020) to achieve $\mathcal{W}(\cdot)$ and employed Depth Anything (?) to estimate the maps of the input views (*See the Appendix F for more results with different depth estimation methods*). We simultaneously rendered 24 novel views and set the reverse steps as 100 for high-quality sample generation. For the implementation of Eq. (11), since applying directly weighted sum usually results in blurry, we ordered the feature pixels by the $\|\boldsymbol{\mu}_{t,\mathbf{p}_i} - \widehat{\mathbf{X}}_{0,\mathbf{p}_i}\|_2$ and take the ratio of $\frac{\lambda(t,\mathbf{p}_i)}{1+\lambda(t,\mathbf{p}_i)}$ smaller pixels from $\widehat{\mathbf{X}}_{0,\mathbf{p}_{0i}}$ and the others from $\boldsymbol{\mu}_{t,\mathbf{p}_i}$ to modulate $\widetilde{\boldsymbol{\mu}}_{t,\mathbf{p}_i}$. We choose the values $(v_1, v_2, v_3)$ as $(1e^{-6}, 9e^{-1}, 5e^{-2})$, $(1e^{-6}, 7e^{-1}, 1e^{-2})$, and $(1e^{-6}, 1.75, 3e^{-2})$ for single, sparse, dynamic scene view synthesis.

**Metrics.** We utilized four metrics to measure the reconstruction performance, i.e., *Fréchet Inception Distance (FID)* (Heusel et al., 2017) evaluating the quality and diversity of synthesized views; *Absolute trajectory error (ATE)* (Goel et al., 1999) measuring the difference between the estimated trajectory of a camera or robot and the ground truth trajectory; *Relative pose error (RPE)* (Goel et al., 1999) measuring the drift, where we separately calculated the transition and rotation as RPE-T and RPE-R, respectively. We utilize Particle-SFM (Zhao et al., 2022) to estimate the camera trajectory and assess the pose metrics. Since current depth estimation algorithms struggle to derive absolute depth from a single view or monocular video, resulting in a scale gap between the synthesized and ground truth images, we only compare the paired metrics in Appendix I, such as *LPIPS*.

### 5.1 RESULTS OF NVS FROM SINGLE OR MULTIPLE VIEWS OF STATIC SCENES

Fig. 3 visualizes synthesized novel views of two scenes by different methods from single views, where we can see that our method can consistently generate high-quality novel views with visually pleasing geometry and textures. The quantitative results listed in Table 1 also validate the significant superiority of our method over state-of-the-art methods. Although MontionCtrl (Wang et al., 2024) can generate high-quality images reflected by the FID value, its results have significantly large view pose errors reflected by the higher ATE and RPE values. For Text2Nerf (Zhang et al., 2024), only 7 out of the total 9 sequences can be calculated metrics, which may be induced by the inherent errors of the learned geometric structure.

Fig. 4 shows the synthesized views by different methods from two given views, where it can be seen that our method outperforms state-of-the-art methods by clearer views, especially for the $2^{nd}$ and $3^{rd}$

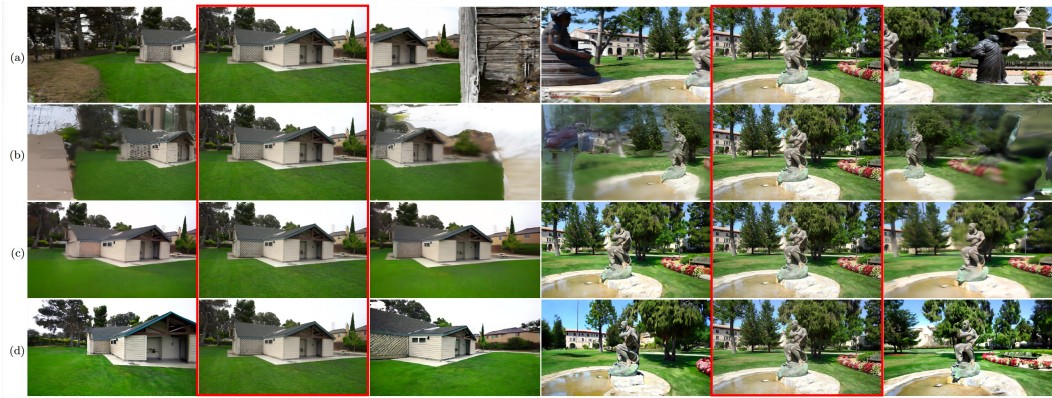

Figure 3: Visual comparison of single view-based NVS results by (**a**) Text2Nerf (Zhang et al., 2024), (**b**) 3D-aware (Xiang et al., 2023), (**c**) MotionCtrl (Wang et al., 2024), (**d**) Ours (Post). The middle view of each scene highlighted with the red rectangle refers to the input view. Here, we only show the results of the best two of all compared methods. *We also refer reviewers to the **Appendix B** and **video demo** contained in the supplementary file for more impressive results and comparisons.*

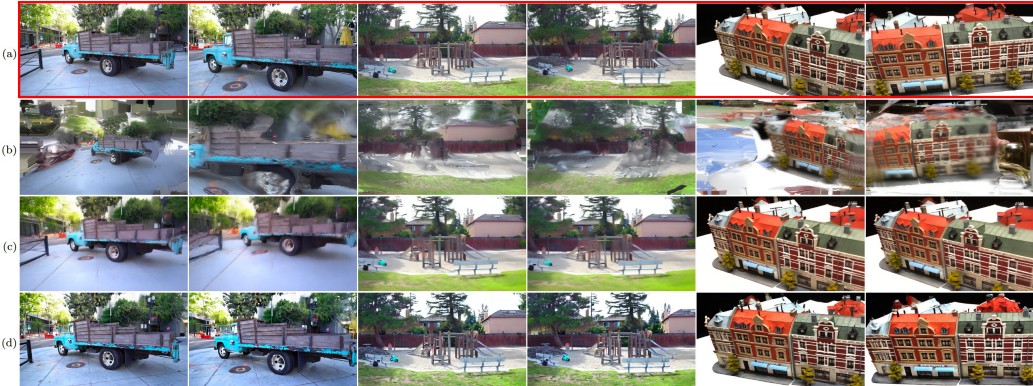

Figure 4: (**a**) The two input views of each scene highlighted with the red rectangle. Visual results of multiview-based NVS by (**b**) 3D-aware (Xiang et al., 2023), (**c**) MotionCtrl (Wang et al., 2024), (**d**) Ours (Post).

scenes. In addition, it is worth noting that the lower ATE of 3D-aware (Xiang et al., 2023) is due to the incomplete evaluation (*See the Appendix C for more results*).

Note that our method can also achieve 360° NVS through iterative execution of the proposed algorithm. Fig. 5 shows the synthesized 360° NVS from both single-view and multi-view inputs, demonstrating the capability of the proposed method to effectively handle the NVS task (*See the Appendix H for the detailed strategy*).

## 5.2 RESULTS OF NVS FROM MONOCULAR VIDEOS OF DYNAMIC SCENES

The non-generative Gaussian-based methods Deformable-Gaussian (Yang et al., 2024c) and 4D-Gaussian (Wu et al., 2024a) usually cannot handle the marginal area, as illustrated in $2^{nd}$, $4^{th}$ and $6^{th}$ columns of Fig. 6 (b) and (c). Although the SVD-based method MotionCtrl can generate the boundary region as shown in Fig. 6 (e), the synthesized views are blurry and the poses of generated samples cannot follow

Table 2: Quantitative comparison of different methods on NVS from monocular videos of dynamic scenes. *For all metrics, the lower, the better.*

| Methods | Train | FID | ATE | RPE-T | RPE-R |
|---|---|---|---|---|---|
| Deformable-Gaussian (Yang et al., 2024c) | ✓ | 115.82 | 1.813 * | 0.678 * | 0.613 * |
| 4D-Gaussian (Wu et al., 2024a) | ✓ | 74.34 | 2.087 | 0.625 | 0.825 |
| 3D-aware (Xiang et al., 2023) | × | 159.03 | 3.100 | 1.343 | 1.368 |
| MotionCtrl (Wang et al., 2024) | × | 70.35 | 3.384 * | 1.069 * | 0.653 * |
| **Ours (DGS)** | × | 37.973 | 2.236 | 0.691 | 0.446 |
| **Ours (Post)** | × | 39.86 | 2.308 | 0.725 | 0.400 |

the prompt, especially on the $3^{rd}$ sample. However, our method consistently generates high-quality novel views with lower pose errors, indicating its strong potential.

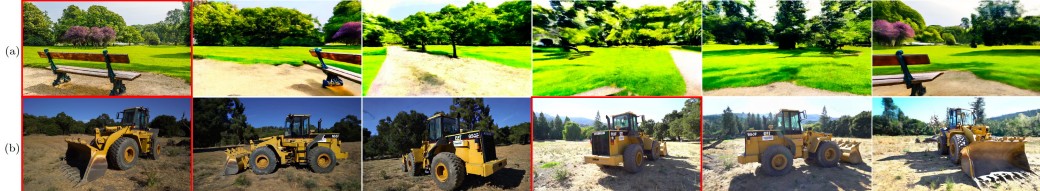

Figure 5: The input views of each scene highlighted with the red rectangle. Visual results of synthesized 360° NVS from (**a**) single view and (**b**) multi-view input.

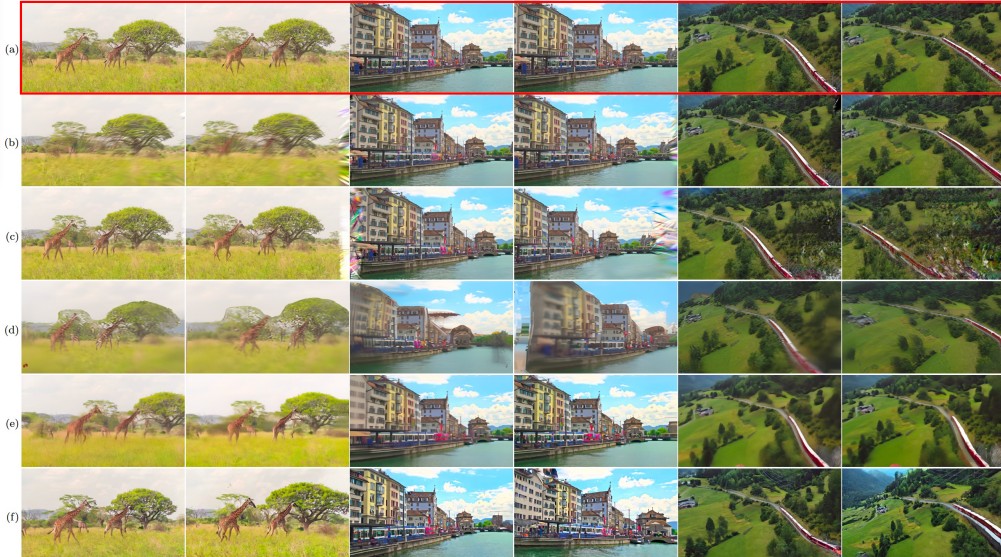

Figure 6: Visual comparison on dynamic scene view synthesis of (**a**) input frames in the corresponding time of generated images, (**b**) Deformable-Gaussian (Yang et al., 2024c), (**c**) 4D-Gaussian (Wu et al., 2024a), (**d**) 3D-aware (Xiang et al., 2023), (**e**) MotionCtrl (Wang et al., 2024), (**f**) Ours (Post).

## 5.3 ABLATION STUDY

**Reverse Inference Steps**. The quantitative results in Table 3 show that the synthesized image quality of our method does not improve intensely with the number of inference steps increasing. However, the pose error decreases significantly, indicating the necessity of a sufficient number of inference steps for accurately rendering novel views.

**Posterior Sampling *vs*. Directly Guided Sampling**. The quantitative results listed in Tables 1 and 2 show that both the Ours (DGS) and Ours (Post) can generate high-quality images with comparable FID. However, the view pose of Ours (Post) is much more accurate than Ours (DGS), which is also visually verified by the results in Figs. 7 (c) and (d). Besides, Ours (DGS) takes 6 minutes to render 25 views, while Ours (Post) uses 1 hour.

Table 3: Quantitative comparison of different numbers of inference steps. *For all metrics, the lower, the better.*

| Inference step | FID | ATE | RPE-T | RPE-R |
|---|---|---|---|---|
| 25 | 175.432 | 5.317 | 0.691 | 0.847 |
| 50 | 168.938 | 2.275 | 0.428 | 0.402 |
| 100 | 165.12 | 0.767 | 0.156 | 0.170 |

**Effectiveness of Back-propagation in Posterior Sampling**. We performed ablation studies on the updating rate $\kappa$ and the normalization schemes in Eq. (14). The experimental results shown in Tab. 4 empirically guarantee that the back-propagation brings the latent space closer to the inherent low-dimensional manifold structure during each update step, further validating the performance of the proposed score-modulation schemes.

Table 4: Quantitative comparison of ablating updating rate $\kappa$ and the normalization schemes on Ours (Post). *For all metrics, the lower, the better.*

| $\kappa$ | Normalization | FID | ATE | RPE-T | RPE-R |
|---|---|---|---|---|---|
| $2e^{-2}\sqrt{\sigma(t)}$ | Y | 165.12 | 0.767 | 0.156 | 0.170 |
| 0.5 | Y | 268.17 | 1.92 | 0.318 | 0.399 |
| $2e^{-2}\sqrt{\sigma(t)}$ | N | 176.00 | 4.71 | 0.568 | 1.05 |

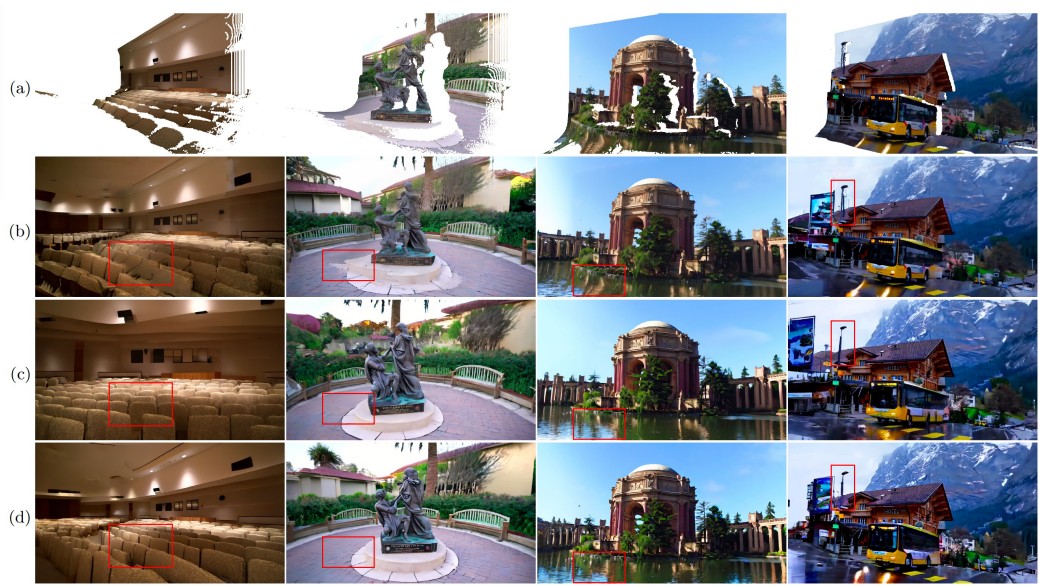

Figure 7: Ablation studies of the proposed weight strategy of $\widehat{\lambda}(t, \mathbf{p}_i)$ and embedding strategy. Visual comparison of (**a**) warped input image $\widehat{\mathbf{X}}_{0,\mathbf{p}_i}$, (**b**) results without our proposed weight strategy, (**c**) results of Ours (DGS), and (**d**) results of Ours (Post).

**Effectiveness of Adaptive** $\widehat{\lambda}(t, \mathbf{p}_i)$. Here, we illustrate the effectiveness of adaptively adjusting $\widehat{\lambda}(t, \mathbf{p}_i)$ via setting it to $+\infty$, i.e., $\widetilde{\boldsymbol{\mu}}_{t,\mathbf{p}_i} = \widehat{\mathbf{X}}_{0,\mathbf{p}_i}$. Visual comparison results in Fig. 7 indicates that the proposed method significantly correct *the warping errors* ($1^{st}$, $2^{nd}$ and $4^{th}$ samples in Fig. 7) and *non-Lambert reflection* ($3^{rd}$ sample in Fig. 7) as indicated in $\mathcal{E}_T$ of Eq. (10).

**Different Trajectories**. We detailedly visualize the result of the proposed method on different trajectories. We apply transitions in eight different directions. The results are shown in Fig. 8. The proposed method can consistently render high-quality novel views.

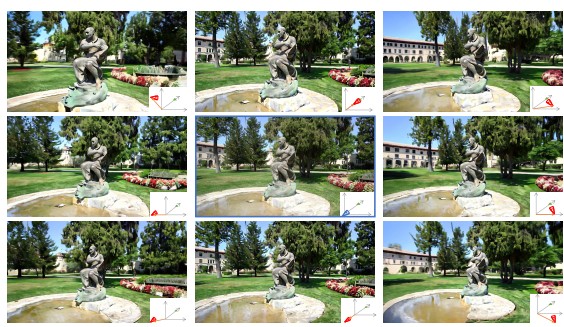

Figure 8: NVS results of Ours (Post) with different trajectories, where the given view is bounded by a blue circle. We draw the pose at the right bottom corner of each sub-figure.

## 6 CONCLUSION & DISCUSSION

We have presented a training-free novel view synthesis paradigm. The proposed method achieves remarkable performance compared with state-of-the-art methods. The advantages are credited to the powerful generative capacity of the pre-trained large stable video diffusion model and our elegant designs of the adaptive modulation of the diffusion score function through comprehensive theoretical and empirical analyses.

Although our method takes longer than existing methods, the promising generative capacity inherent in the large pre-trained diffusion model may attract improvements in the future. Since the proposed method can synthesize views more accurate poses, we believe it may be a potential solution for the distillation of a pose controllable video diffusion model. In the age of generative intelligence, the authors expect the proposed algorithm would inspire future work to unify computer graphics pipelines with generative models. Finally, the authors sincerely appreciate Stability AI for the open-sourced SVD (Blattmann et al., 2023).

ETHICS & REPRODUCIBILITY STATEMENTS

The proposed method is specifically designed for novel view synthesis through video diffusion. As such, no additional information regarding human subjects or potentially harmful insights is introduced during the process. This approach prioritizes privacy and ethical considerations by relying solely on the data available from the input images or videos, without requiring any sensitive or external information. Furthermore, all of our experiments are conducted in a training-free manner, which not only simplifies implementation but also ensures reproducibility. More importantly, the source code can be found on the webpage: https://github.com/ZHU-Zhiyu/NVS_Solver.

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

CONTENTS

## A   SCENE PRIOR-BASED SCORE-MODULATION PRESERVES DATA MANIFOLD

### A.1   MODULATED SCORE IS MORE ACCURATE

In this section, we first illustrate that the proposed modulation is more accurate than guided sampling in an inpainting-like manner. Denote by $\widehat{\mathbf{X}}_{0,p_i}$ the warping content for scene's prior on pose $p_i$. Then, the vanilla guided sampling methods, especially the inpainting-based method, directly utilize $\widehat{\mathbf{X}}_{0,p_i}$ as the guidance prior. Thus, its score estimation error $\mathcal{E}$ is shown as

$$\mathcal{E}_P = \left\| \widehat{\boldsymbol{\mu}}_{t,\mathbf{p}_i} - \widehat{\boldsymbol{\mu}}_{t,\mathbf{p}_i}^* \right\|_2 = \left\| \widehat{\mathbf{X}}_{0,p_i} - \widehat{\boldsymbol{\mu}}_{t,\mathbf{p}_i}^* \right\|_2 = v_3 \left\| \Delta \mathbf{p}_i \right\|_2, \tag{19}$$

where $\Delta \mathbf{p}_i$ indicates the variations of camera pose between the given and target views. However, for our modulated score, its score estimation error $\mathcal{E}_M$ is shown as

$$\mathcal{E}_M = \left\| \frac{1}{1 + \lambda(t, \mathbf{p}_i)} \boldsymbol{\mu}_{t,\mathbf{p}_i} + \frac{\lambda(t, \mathbf{p}_i)}{1 + \lambda(t, \mathbf{p}_i)} \widehat{\mathbf{X}}_{0,\mathbf{p}_i} - \widehat{\boldsymbol{\mu}}^*_{t,\mathbf{p}_i} \right\|_2 \tag{20}$$

$$= \left\| \frac{1}{1 + \lambda(t, \mathbf{p}_i)} (\boldsymbol{\mu}_{t,\mathbf{p}_i} - \widehat{\boldsymbol{\mu}}^*_{t,\mathbf{p}_i}) + \frac{\lambda(t, \mathbf{p}_i)}{1 + \lambda(t, \mathbf{p}_i)} (\widehat{\mathbf{X}}_{0,\mathbf{p}_i} - \widehat{\boldsymbol{\mu}}^*_{t,\mathbf{p}_i}) \right\|_2 \tag{21}$$

$$\leq \frac{1}{1 + \lambda(t, \mathbf{p}_i)} \left\| \boldsymbol{\mu}_{t,\mathbf{p}_i} - \widehat{\boldsymbol{\mu}}^*_{t,\mathbf{p}_i} \right\|_2 + \frac{\lambda(t, \mathbf{p}_i)}{1 + \lambda(t, \mathbf{p}_i)} \left\| \widehat{\mathbf{X}}_{0,\mathbf{p}_i} - \widehat{\boldsymbol{\mu}}^*_{t,\mathbf{p}_i} \right\|_2 \tag{22}$$

$$= \frac{v_2 \sigma(t)}{1 + \lambda(t, \mathbf{p}_i)} + \frac{\lambda(t, \mathbf{p}_i)}{1 + \lambda(t, \mathbf{p}_i)} v_3 \|\Delta \mathbf{p}\|_2 \tag{23}$$

By substituting the utilized solution of $\widehat{\lambda}(t, \mathbf{p}_i)$ from Eq. (18), with $Q = v_3 \|\Delta \mathbf{p}\|_2 - v_2 \sigma(t)$, we can derive that

$$\mathcal{E}_M \leq \frac{2 v_1 v_2 \sigma(t)}{-Q + \sqrt{Q^2 + 4 v_1 Q}} + \frac{-(2 v_1 + Q) + \sqrt{Q^2 + 4 v_1 Q}}{-Q + \sqrt{Q^2 + 4 v_1 Q}} v_3 \|\Delta \mathbf{p}\|_2, \tag{24}$$

$$= \frac{2 v_1 v_2 \sigma(t)}{-Q + \sqrt{Q^2 + 4 v_1 Q}} + \frac{-2 v_1}{-Q + \sqrt{Q^2 + 4 v_1 Q}} v_3 \|\Delta \mathbf{p}\|_2 + \mathcal{E}_P, \tag{25}$$

$$= \frac{-2 v_1 Q}{-Q + \sqrt{Q^2 + 4 v_1 Q}} + \mathcal{E}_P. \tag{26}$$

since $v_1$ is a small value we make further approximation that

$$\mathcal{E}_P \leq \lim_{v_1 \to 0} \frac{-2 v_1 Q}{-Q + \sqrt{Q^2 + 4 v_1 Q}} + \mathcal{E}_P, \tag{27}$$

$$\overset{\textcircled{1}}{\approx} \lim_{v_1 \to 0} \frac{-2 v_1 Q}{-Q + Q + \frac{1}{2\sqrt{Q^2 + 4 v_1 Q}}} + \mathcal{E}_P, \tag{28}$$

$$= \lim_{v_1 \to 0} \frac{-2 v_1 Q}{-Q + Q + \frac{1}{2\sqrt{Q^2 + 4 v_1 Q}}} + \mathcal{E}_P, \tag{29}$$

$$= \lim_{v_1 \to 0} -4 v_1 Q^2 + \mathcal{E}_P, \tag{30}$$

$$\approx \mathcal{E}_P, \tag{31}$$

where $\textcircled{1}$ indicates to apply Taylor series of $\sqrt{Q^2 + 4 v_1 Q}$. Thus, $\mathcal{E}_M \lesssim \mathcal{E}_P$ indicates that the proposed method modulates a more accurate target score function with less error.

## A.2 GUIDED POSTERIOR SAMPLING OF MODULATED SCORE IS ON DATA MANIFOLD

According to Chung et al. (2023); Huang et al. (2022), we define a local tangent space as $\mathcal{T}_{\boldsymbol{x}}\mathcal{M}$ for a local orthogonal projection onto manifold $\mathcal{M}$, with a transition process

$$\mathbf{Q}_t : \mathbb{R}^d \to \mathbb{R}^d, \boldsymbol{x}_t \to \boldsymbol{\mu}_t = \mathcal{X}_{\boldsymbol{\theta}}(\boldsymbol{x}_t, t) \tag{32}$$

$$\frac{\partial}{\partial \boldsymbol{x}} \|\boldsymbol{\mu}_t - \widetilde{\boldsymbol{\mu}}_t\|_2^2 = 2 \frac{\partial \boldsymbol{\mu}_t}{\partial \boldsymbol{x}} (\boldsymbol{\mu}_t - \widetilde{\boldsymbol{\mu}}_t). \tag{33}$$

Since for the Jacobin matrix $\mathcal{J}_{\mathbf{Q}_t} = \frac{\partial \boldsymbol{\mu}_t}{\partial \boldsymbol{x}}$ denotes a transition which maps a vector to the tangent space of function $\mathbf{Q}_t$. Thus, we have

$$\frac{\partial}{\partial \boldsymbol{x}} \|\boldsymbol{\mu}_t - \widetilde{\boldsymbol{\mu}}_t\|_2^2 = \mathcal{J}_{\mathbf{Q}_t} \left( 2(\boldsymbol{\mu}_t - \widetilde{\boldsymbol{\mu}}_t) \right) \in T_{\mathbf{Q}_i}\mathcal{M}, \tag{34}$$

The aforementioned proof indicates that our introduced regularization, especially (DGS), only advocates moving the latent in the orthogonal manifold direction. Moreover, considering that the updating step size is relatively small as $2e^{-2}\sqrt{\sigma(t)}$ compared to the noised latent of $\sqrt{\sigma^2(t)+1}$, we have

$$\frac{2e^{-2}\sqrt{\sigma(t)}}{\sqrt{\sigma^2(t)+1}} = \frac{2e^{-2}}{\sqrt{\sigma(t)+\frac{1}{\sigma(t)}}} \leq \frac{2e^{-2}}{\sqrt{2}} = \sqrt{2}e^{-2}. \tag{35}$$

Thus, both the orthogonal regularization direction and small updating step make the proposed method to be safely moving on the data manifold.

## B  VISUAL COMPARISON DETAILS

In this section, we visually demonstrate the results as shown in Figs. B-1, B-2, and B-3. The visual result demonstrates the superiority of the proposed method.

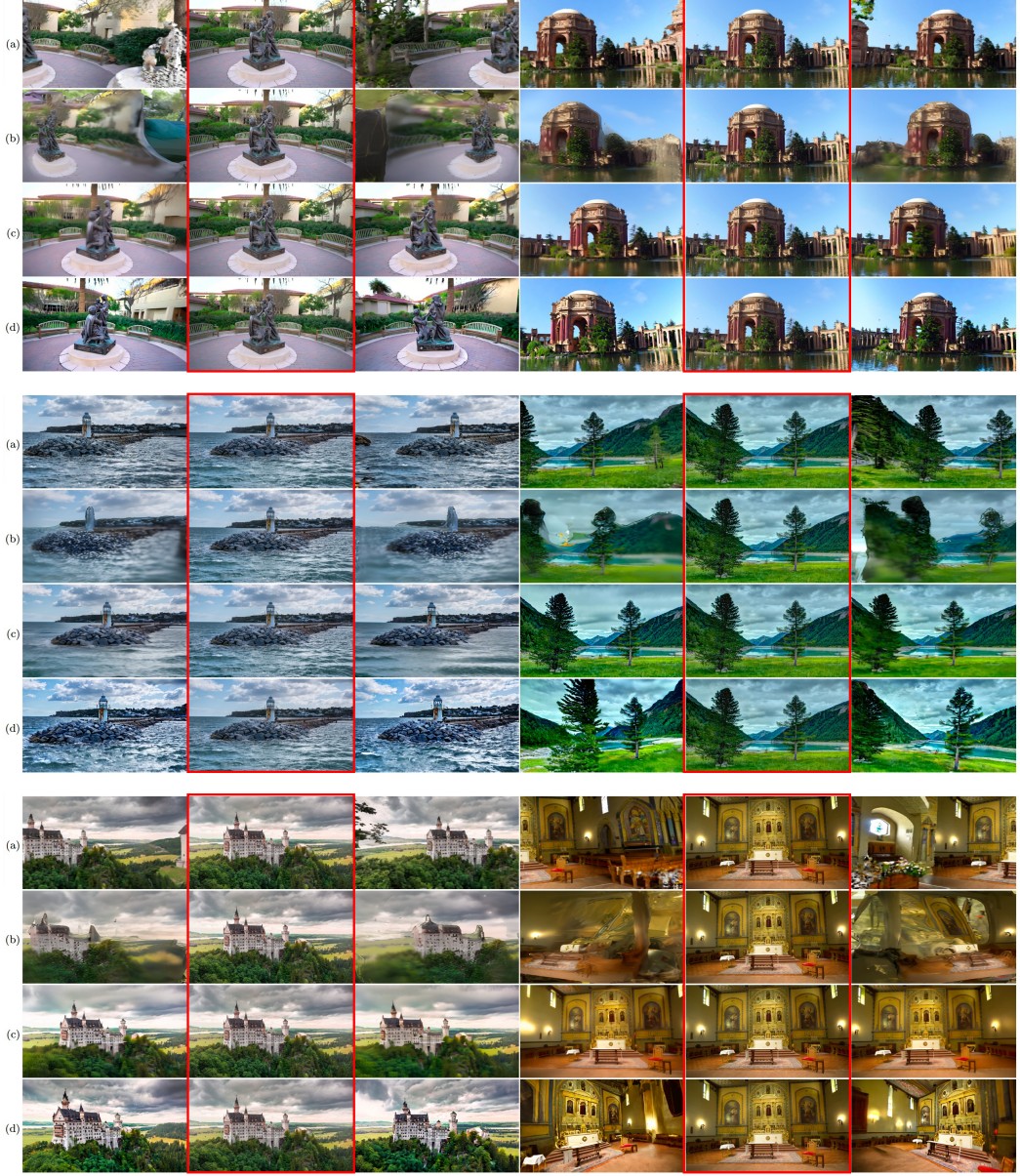

Figure B-1: Visual comparison of single view-based NVS results by (a) Text2Nerf (Zhang et al., 2024), (b) 3D-aware (Xiang et al., 2023), (c) MotionCtrl (Wang et al., 2024), (d) Ours (Post). The middle view of each scene highlighted with the red rectangle refers to the input view.

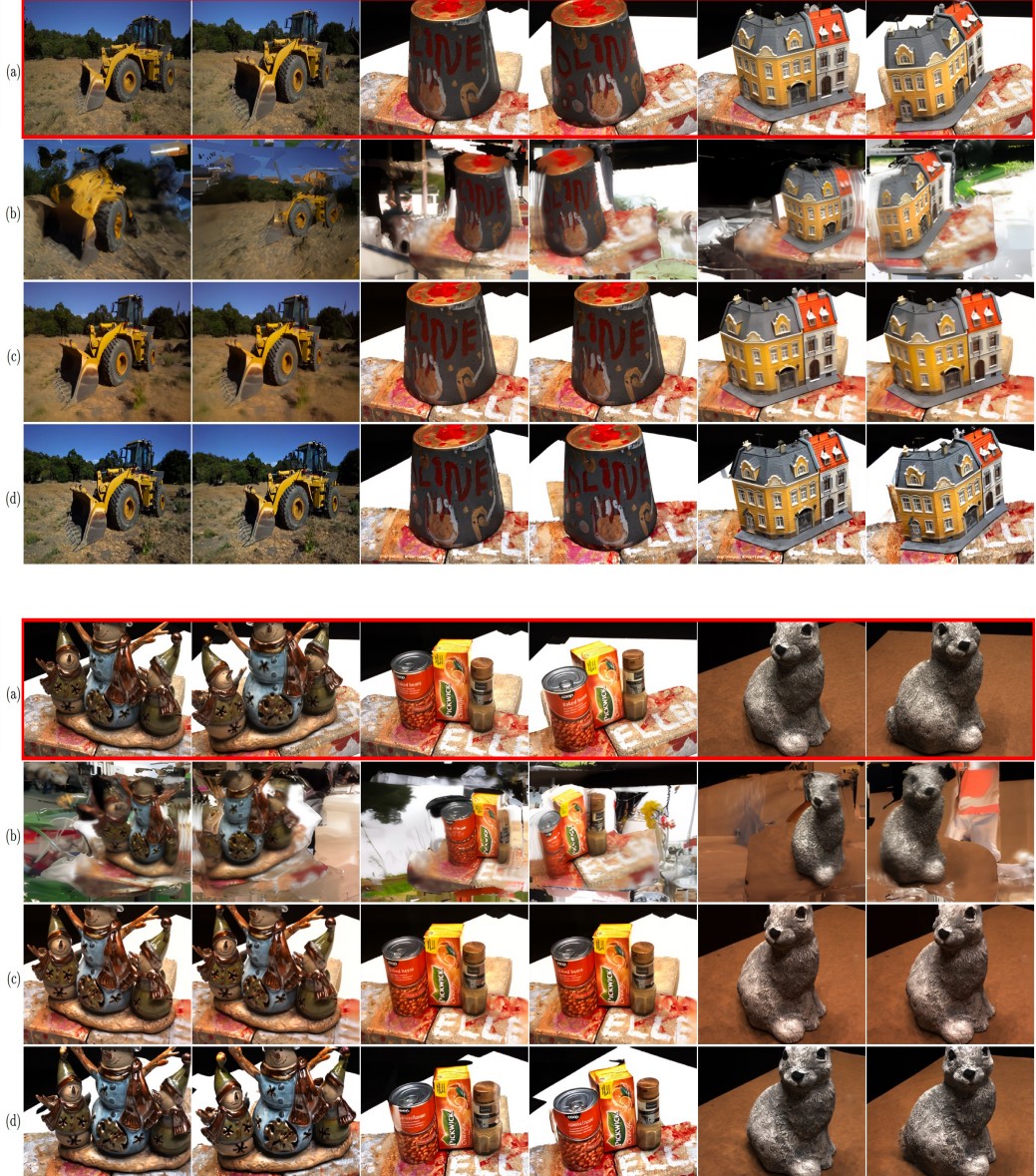

Figure B-2: (**a**) The two input views of each scene highlighted with the red rectangle. Visual results of multiview-based NVS by (**b**) 3D-aware (Xiang et al., 2023), (**c**) MotionCtrl (Wang et al., 2024), (**d**) Ours (Post).

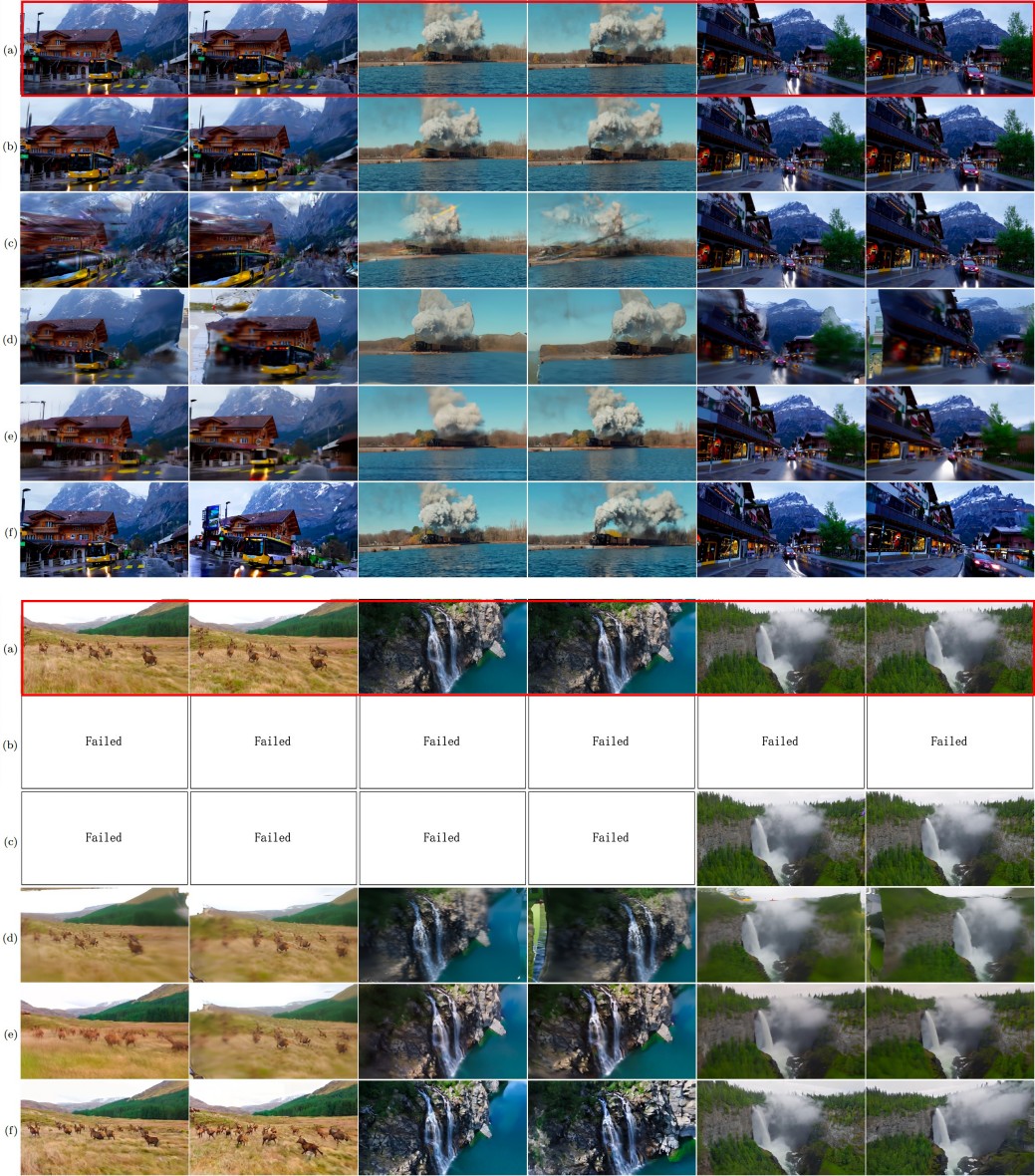

Figure B-3: Visual comparison on dynamic scene view synthesis of (**a**) input frames in the corresponding time of generated images, (**b**) Deformable-Gaussian (Yang et al., 2024c), (**c**) 4D-Gaussian (Wu et al., 2024a), (**d**) 3D-aware (Xiang et al., 2023), (**e**) MotionCtrl (Wang et al., 2024), (**f**) Ours (Post). 'Failed' refers to the method cannot work on the condition.

## C    QUANTITATIVE COMPARISON DETAILS

We give detailed quantitative comparisons of different methods as shown in Tables C-1, C-2 and C-3. The results demonstrate that the proposed methods, Ours (DGS) and (Post) outperform SOTA methods in most scenes.

Table C-1: Quantitative comparison of different methods on single view synthesis. *For all metrics, the lower, the better.*

| Scene | Metric | Sparse Gaussian | Sparse Nerf | Text2Nerf | 3D-aware | MotionCtrl | **Ours (DGS)** | **Ours (Post)** |
|---|---|---|---|---|---|---|---|---|
| | ATE | – – | – – | 4.424 | 2.850 | 7.111 | 17.160 | 1.261 |
| Auditorium | RPE-T | – – | – – | 2.020 | 3.398 | 1.448 | 2.669 | 0.265 |
| | RPE-R | – – | – – | 0.266 | 2.609 | 1.320 | 1.192 | 0.144 |
| | ATE | – – | – – | 1.133 | 2.832 | 3.217 | 2.638 | 0.430 |
| Barn | RPE-T | – – | – – | 0.307 | 0.868 | 0.369 | 0.533 | 0.104 |
| | RPE-R | – – | – – | 0.064 | 2.336 | 1.074 | 0.632 | 0.197 |
| | ATE | – – | – – | – – | 2.831 | 3.045 | 3.038 | 0.883 |
| Castle | RPE-T | – – | – – | – – | 1.425 | 0.716 | 0.634 | 0.171 |
| | RPE-R | – – | – – | – – | 1.036 | 0.483 | 0.983 | 0.039 |
| | ATE | – – | – – | 2.167 | 2.830 | 3.256 | 3.017 | 0.664 |
| Church | RPE-T | – – | – – | 0.486 | 0.860 | 0.798 | 0.636 | 0.107 |
| | RPE-R | – – | – – | 0.093 | 2.476 | 1.553 | 1.000 | 0.176 |
| | ATE | – – | – – | 1.264 | 2.830 | 1.941 | 0.944 | 1.023 |
| Family | RPE-T | – – | – – | 0.377 | 0.867 | 0.378 | 0.212 | 0.191 |
| | RPE-R | – – | – – | 0.050 | 2.295 | 0.890 | 0.461 | 0.514 |
| | ATE | – – | – – | 1.561 | 2.831 | 3.431 | 1.226 | 0.408 |
| Ignatius | RPE-T | – – | – – | 0.365 | 0.866 | 0.495 | 0.342 | 0.129 |
| | RPE-R | – – | – – | 0.050 | 2.298 | 1.043 | 0.894 | 0.243 |
| | ATE | – – | – – | – – | 2.835 | 3.903 | 0.959 | 1.062 |
| Palace | RPE-T | – – | – – | – – | 0.904 | 0.517 | 0.221 | 0.192 |
| | RPE-R | – – | – – | – – | 0.578 | 0.363 | 0.313 | 0.101 |
| | ATE | – – | – – | 2.385 | 2.830 | 4.007 | 5.292 | 0.395 |
| Seaside | RPE-T | – – | – – | 0.754 | 0.717 | 0.522 | 0.639 | 0.090 |
| | RPE-R | – – | – – | 0.073 | 0.473 | 0.283 | 0.311 | 0.040 |
| | ATE | – – | – – | 2.628 | 2.854 | 4.740 | 6.521 | 0.776 |
| Trees | RPE-T | – – | – – | 0.718 | 1.420 | 1.098 | 1.399 | 0.155 |
| | RPE-R | – – | – – | 0.151 | 0.859 | 0.506 | 0.894 | 0.079 |

Table C-2: Quantitative comparison of different methods on sparse view synthesis. *For all metrics, the lower, the better.*

| Scene | Metric | Sparse Gaussian | Sparse Nerf | Text2Nerf | 3D-aware | MotionCtrl | **Ours (DGS)** | **Ours (Post)** |
|---|---|---|---|---|---|---|---|---|
| | ATE | – – | 1.697 | – – | – – | 2.826 | 2.250 | 0.330 |
| caterpillar | RPE-T | – – | 0.046 | – – | – – | 0.040 | 0.010 | 0.002 |
| | RPE-R | – – | 2.337 | – – | – – | 0.680 | 0.305 | 0.032 |
| | ATE | – – | 2.813 | – – | 0.721 | 1.918 | 0.597 | 0.033 |
| playground | RPE-T | – – | 0.058 | – – | 0.059 | 0.029 | 0.008 | 0.001 |
| | RPE-R | – – | 0.156 | – – | 3.534 | 0.524 | 0.201 | 0.113 |
| | ATE | – – | 2.556 | – – | – – | 2.718 | 0.058 | 0.068 |
| truck | RPE-T | – – | 0.142 | – – | – – | 0.088 | 0.010 | 0.002 |
| | RPE-R | – – | 11.96 | – – | – – | 0.954 | 0.287 | 0.048 |
| | ATE | – – | 7.980 | – – | 2.872 | – – | 33.29 | 7.069 |
| scan1 | RPE-T | – – | 2.518 | – – | 8.477 | – – | 31.02 | 3.918 |
| | RPE-R | – – | 0.302 | – – | 1.411 | – – | 2.995 | 0.592 |
| | ATE | – – | 8.318 | – – | – – | 48.00 | 6.556 | 4.480 |
| scan2 | RPE-T | – – | 2.609 | – – | – – | 31.60 | 5.902 | 2.211 |
| | RPE-R | – – | 0.307 | – – | – – | 1.614 | 0.648 | 0.324 |
| | ATE | – – | 8.795 | – – | – – | 46.22 | 8.660 | 7.617 |
| scan3 | RPE-T | – – | 2.633 | – – | – – | 31.20 | 6.104 | 3.976 |
| | RPE-R | – – | 0.269 | – – | – – | 1.656 | 0.940 | 0.564 |
| | ATE | – – | 7.902 | – – | 2.886 | 53.38 | 59.03 | 3.436 |
| scan5 | RPE-T | – – | 2.510 | – – | 8.495 | 16.13 | 32.65 | 2.270 |
| | RPE-R | – – | 0.297 | – – | 1.413 | 1.526 | 2.551 | 0.326 |
| | ATE | – | 7.072 | – – | – – | 48.75 | 55.68 | 6.504 |
| scan15 | RPE-T | – | 2.394 | – – | – – | 30.57 | 42.98 | 2.796 |
| | RPE-R | – | 0.327 | – – | – – | 1.692 | 1.690 | 0.352 |
| | ATE | – | 8.025 | – – | – – | 97.60 | 31.86 | 6.936 |
| scan55 | RPE-T | – | 2.487 | – – | – – | 47.21 | 37.41 | 3.098 |
| | RPE-R | – | 0.275 | – – | – – | 4.520 | 2.422 | 0.618 |

Table C-3: Quantitative comparison of different methods on dynamic view synthesis. '_1/_2' means that we synthesize novel views by moving the camera on the right/left circle trajectory. *For all metrics, the lower, the better*.

| Scene | Metric | Deformable-Gaussian | 4D-Gaussian | 3D-aware | MotionCtrl | **Ours (DGS)** | **Ours (Post)** |
|---|---|---|---|---|---|---|---|
| Street_1 | ATE | 2.834 | 2.817 | 2.847 | 2.746 | 0.619 | 0.619 |
| | RPE-T | 1.463 | 1.342 | 1.383 | 0.811 | 0.306 | 0.307 |
| | RPE-R | 1.009 | 1.322 | 1.0340 | 0.906 | 0.217 | 0.218 |
| Street_2 | ATE | 0.486 | 0.819 | 1.470 | – – | 2.842 | 2.842 |
| | RPE-T | 0.169 | 0.163 | 0.235 | – – | 0.647 | 0.647 |
| | RPE-R | 0.439 | 0.800 | 0.528 | – – | 0.604 | 0.602 |
| Kangaroo_1 | ATE | – – | 3.546 | 4.193 | 1.396 | 3.421 | 3.131 |
| | RPE-T | – – | 0.979 | 1.347 | 0.778 | 1.501 | 1.482 |
| | RPE-R | – – | 0.361 | 2.637 | 0.606 | 0.574 | 0.444 |
| Kangaroo_2 | ATE | 3.302 | 2.438 | 4.099 | 5.681 | 5.922 | 4.248 |
| | RPE-T | 1.672 | 0.852 | 2.489 | 1.973 | 2.065 | 1.608 |
| | RPE-R | 0.639 | 0.372 | 0.954 | 0.303 | 0.791 | 0.353 |
| Train3_1 | ATE | 1.658 | 1.187 | 4.667 | 5.626 | 3.744 | 1.130 |
| | RPE-T | 0.698 | 0.650 | 2.642 | 1.501 | 0.999 | 0.402 |
| | RPE-R | 0.468 | 0.588 | 1.494 | 0.471 | 0.455 | 0.245 |
| Train3_2 | ATE | 1.144 | 2.880 | 4.031 | 2.186 | 1.026 | 1.315 |
| | RPE-T | 0.968 | 1.108 | 1.748 | 0.908 | 0.448 | 0.476 |
| | RPE-R | 0.785 | 1.483 | 2.266 | 0.574 | 0.567 | 0.529 |
| Train5_1 | ATE | 2.838 | 1.600 | 3.011 | 3.274 | 0.624 | 0.815 |
| | RPE-T | 0.313 | 0.252 | 1.537 | 0.880 | 0.196 | 0.165 |
| | RPE-R | 0.546 | 0.699 | 1.523 | 1.042 | 0.453 | 0.629 |
| Train5_2 | ATE | 1.046 | 2.3516 | 2.708 | 3.3222 | 0.851 | 2.874 |
| | RPE-T | 0.335 | 0.377 | 0.513 | 0.719 | 0.1916 | 0.991 |
| | RPE-R | 0.734 | 1.686 | 1.347 | 0.666 | 0.599 | 0.653 |
| Road_1 | ATE | 0.521 | 0.548 | 1.195 | 2.783 | 0.774 | 3.328 |
| | RPE-T | 0.219 | 0.251 | 0.667 | 1.061 | 0.326 | 1.016 |
| | RPE-R | 0.330 | 0.363 | 0.679 | 0.479 | 0.100 | 0.239 |
| Road_2 | ATE | 2.492 | 2.679 | 2.775 | 3.445 | 2.534 | 2.781 |
| | RPE-T | 0.260 | 0.275 | 0.867 | 0.991 | 0.230 | 0.158 |
| | RPE-R | 0.564 | 0.579 | 1.214 | 0.828 | 0.099 | 0.091 |

# D   VISUALIZATION OF $\widehat{\lambda}(t, \mathbf{p}_i)$

In Fig. D-4, we visualize the $\widehat{\lambda}(t, \mathbf{p}_i)$ of certain sequences in different NVS conditions.

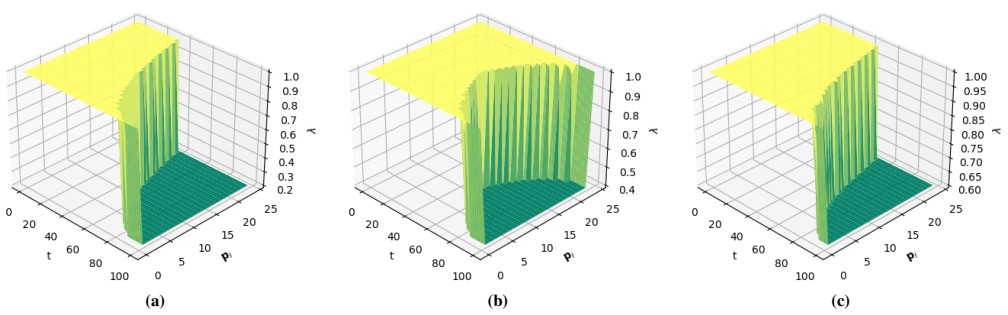

Figure D-4: Visualization of $\widehat{\lambda}(t, \mathbf{p}_i)$ for (a) single view synthesis, (b) sparse view synthesis (c) dynamic view synthesis.

# E   WARPING STRATEGY

We illustrated the warping strategies for different NVS conditions in Fig. E-5.

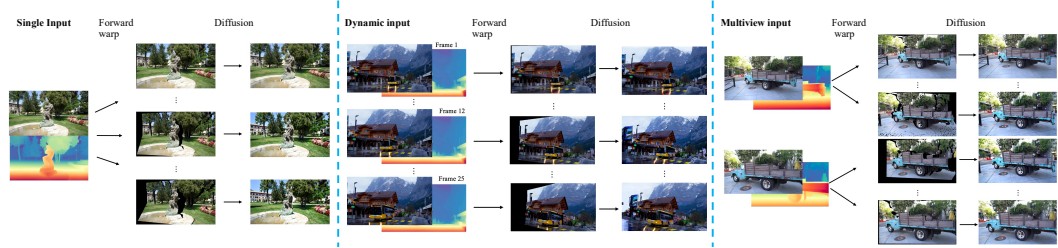

Figure E-5: Illustration of different warping patterns. From left to right, there are single-view, monocular video, and multi-view-based NVS.

## F   PERFORMANCE OF OUR METHOD ACROSS DIFFERENT DEPTH ESTIMATION METHODS

We conducted experiments on our method using various depth estimation approaches, including DINOV2 (Oquab et al., 2023) and DepthAnything V2 (Yang et al., 2024b). The results, shown in Table F-4, clearly demonstrate that our method consistently performs well across different depth estimation techniques, surpassing the state-of-the-art NVS method, MotionCtrl (Wang et al., 2024). As expected, better depth maps indeed lead to more accurate NVS, as evidenced by the comparison between DepthAnythingV1 (Yang et al., 2024a) and DINOV2 Oquab et al. (2023). Furthermore, the performance of DepthAnythingV2 is only comparable to DepthAnythingV1 due to one scene that appears to be an outlier. Upon removing this scene (marked with *), DepthAnythingV2 significantly outperforms DepthAnythingV1. Additionally, DINOV2 achieves a lower FID due to the amplification of rendering pose errors, which facilitates easier reconstruction. Fig. F-6 visualizes the results of our method with DepthAnything V1, V2, and DINOV2. These comparisons confirm the robustness of our method with respect to depth estimation techniques, consistently delivering high performance across different depth estimation modules, including DINOV2, DepthAnything V1, and V2.

Table F-4: Quantitative comparison of our method with DepthAnythingv1  DepthAnythingv2, and DinoV2.*For all metrics, the lower, the better.*

| Methods | FID | ATE | RPE-T | RPE-R |
|---|---|---|---|---|
| MotionCtrl (SOTA) | 179.24 | 3.851 | 0.705 | 0.835 |
| Ours + DepthAnythingv1 | 165.12 | 0.767 | 0.156 | 0.170 |
| Ours + DepthAnythingv2 | 162.896 | 0.831 | 0.110 | 0.170 |
| Ours + DepthAnythingv2* | 163.93 | 0.243 | 0.056 | 0.056 |
| Ours + DinoV2 | 158.24 | 2.395 | 0.454 | 0.674 |

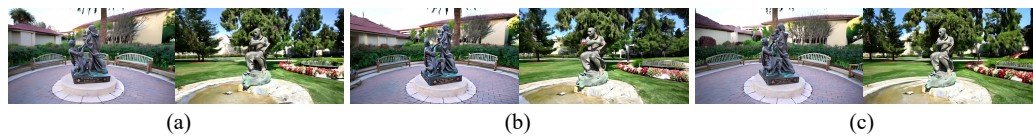

(a)                              (b)                              (c)

Figure F-6: Visual results of our method with (a) DepthAnythingv1, (b) DepthAnythingv2,(c) DinoV2.

## G   COMPARISONS WITH IMAGE INPAINTING RESULTS

Fig. G-7 visualizes the comparison between using an image inpainting method and our approach. Here, we use SDEdit (Meng et al., 2022) as the image inpainting method. When per-view inpainting is used to render novel views, it becomes difficult to ensure that the synthesized views are naturally consistent. As expected, the visual comparisons show that consistency across different views cannot be guaranteed when solving NVS as a per-view inpainting task with SDEdit.

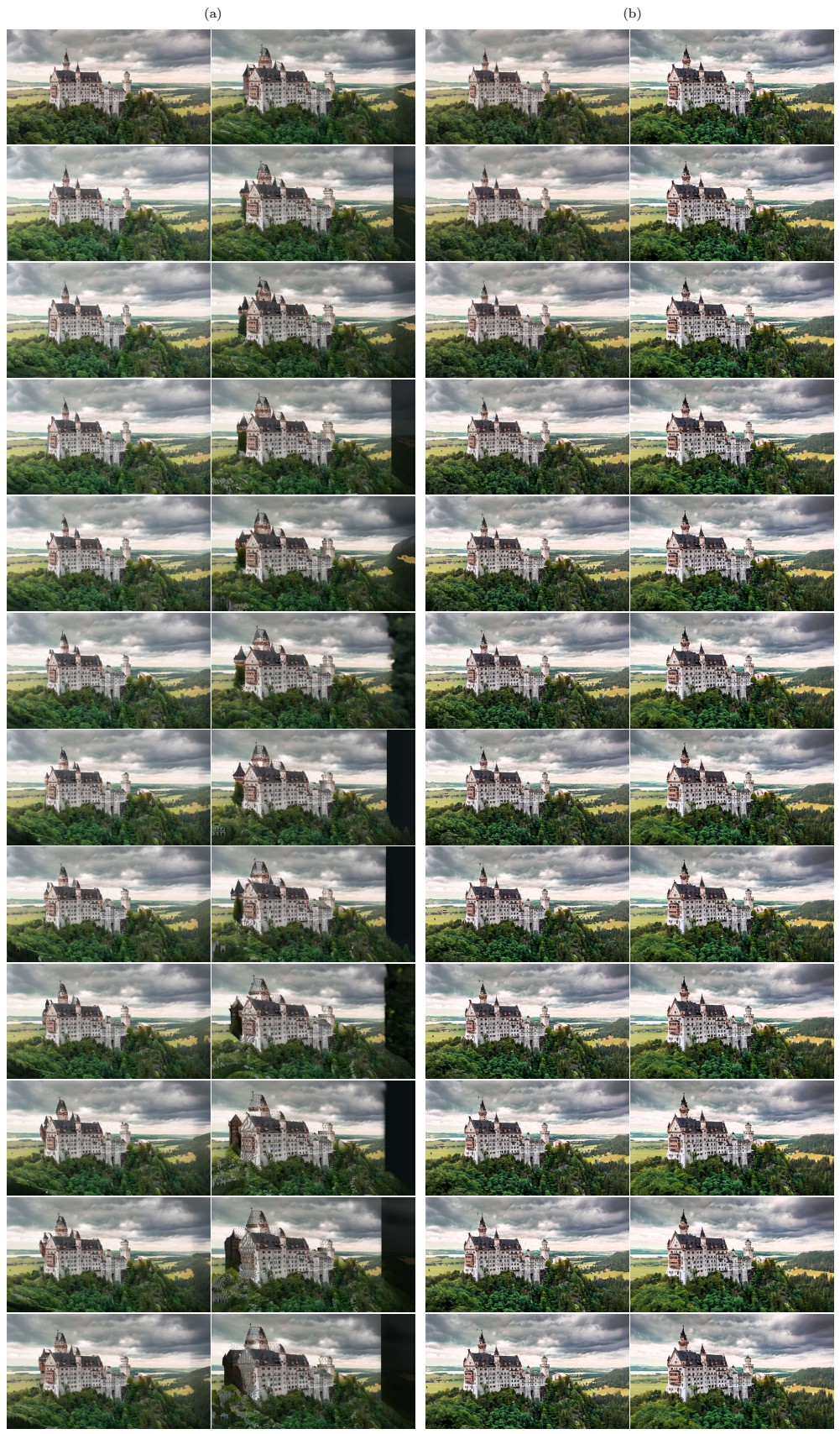

Figure G-7: Visual comparisons of a synthesized video sequence by (a) image inpainting method, and (b) our method.

## H 360° NVS STRATEGY

For a single view input, we first apply the proposed method to rotate 120° to the left and right sides. Moreover, to achieve consistency from different directions, we further process the remaining 120° in a multi-view prompt manner, i.e., we warp both side views to the central as prompt images. Note that in the first two 120° rendering processes, it's also difficult to directly achieve such a large range NVS. Thus, we first reconstruct 30° NVS with 24 frames. Then, we sample 12 frames from the reconstructions as the first 12 prompt images for 60° NVS. Next, we can achieve 24 frames of 120° NVS by sampling 12 prompts from 60° NVS reconstruction. When given multiple views, the proposed method achieves 360° NVS by treating two neighboring images as side views of a pair, warping them towards the central as prompt images.

## I QUANTITATIVE COMPARISON ON LPIPS

We give detailed quantitative comparisons on *LPIPS* of different methods as shown in Tables I-5. The results demonstrate that the proposed methods, Ours (DGS) and (Post) outperform SOTA methods in most scenes.

Table I-5: Quantitative comparison on *LPIPS* of different methods on single view synthesis. *For all metrics, the lower, the better.*

| Scene | Metric | Sparse Gaussian | Sparse Nerf | Text2Nerf | 3D-aware | MotionCtrl | **Ours (DGS)** | **Ours (Post)** |
|---|---|---|---|---|---|---|---|---|
| Auditorium | LPIPS | – – | – – | 0.622 | 0.707 | 0.598 | 0.563 | 0.567 |
| Barn | LPIPS | – – | – – | 0.428 | 0.643 | 0.443 | 0.427 | 0.438 |
| Church | LPIPS | – – | – – | 0.642 | 0.714 | 0.627 | 0.623 | 0.634 |
| Family | LPIPS | – – | – – | 0.647 | 0.794 | 0.540 | 0.523 | 0.525 |
| Ignatius | LPIPS | – – | – – | 0.634 | 0.853 | 0.586 | 0.542 | 0.536 |
| Palace | LPIPS | – – | – – | 0.765 | 0.627 | 0.577 | 0.554 | 0.549 |

## J MESH RECONSTRUCTION

We have applied 2D Gaussian Splatting to reconstruct the mesh on the generated 360° scene. As demonstrated in Fig.J-8, our method successfully maintains geometric consistency across the generated images.

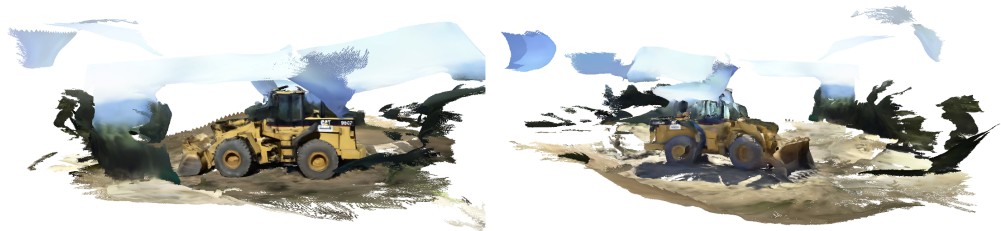

Figure J-8: Mesh reconstruction of synthesized 360° NVS.

## K COMPARISON RESULT ON DYCHECK

We conducted comparison experiments on three scenes from the Dycheck dataset. Table. K-6 illustrates the results of our method alongside two Gaussian-based methods. The three scenes from Dycheck were captured using two cameras, each positioned at a considerable distance from the other. In our experiments, we used a monocular video from one camera as input and generated a video following the trajectory of the other camera. This setup introduces a significant content unoverlap challenge, as the input and target videos capture substantially different perspectives of the scene, making the task particularly difficult. To address potential scale inconsistencies, we utilized the depth maps provided in the dataset. Our method outperformed the 4D-Gaussian approach across all

three metrics.The Deformable-Gaussian method failed to produce viable results in this challenging scenario.

Table K-6: Quantitative comparison of our method with 4D-Gaussian, Deformable-Gaussian on Dycheck. ↑ *(resp. ↓) means the larger (resp. smaller), the better.*

| Methods | PSNR ↑ | SSIM ↑ | LPIPS ↓ |
|---|---|---|---|
| Deformable-Gaussian | – | – | – |
| 4D-Gaussian | 12.68 | 0.346 | 0.737 |
| Ours | 15.84 | 0.385 | 0.410 |

## L    ABLATION ON DIFFERENT WEIGHT FUNCTION

We conducted ablation experiments to evaluate different weight function choices, including (1) a constant value of 0.5

$$\widehat{\lambda}(t, \mathbf{p}_i) = 0.5, \tag{36}$$

(2) a linear function

$$\widehat{\lambda}(t, \mathbf{p}_i) = t, \tag{37}$$

(3) an exponential function

$$\widehat{\lambda}(t, \mathbf{p}_i) = e^{(t+1)}. \tag{38}$$

The results are presented in Table. L-7 and Figure. L-9. These results highlight the superiority of our weight design, as all three simple weight functions lead to decreased performance in generated image quality and trajectory accuracy, particularly when using a constant value for weighting.

Table L-7: Quantitative comparison of different weight functions. *For all metrics, the lower, the better.*

| Methods | FID | ATE | RPE-T | RPE-R |
|---|---|---|---|---|
| Constant (0.5) | 175.68 | 6.09 | 1.06 | 0.864 |
| Linear | 174.23 | 1.04 | 0.210 | 0.199 |
| Exponential | 174.60 | 1.363 | 0.312 | 0.330 |
| Ours | 165.12 | 0.767 | 0.156 | 0.170 |

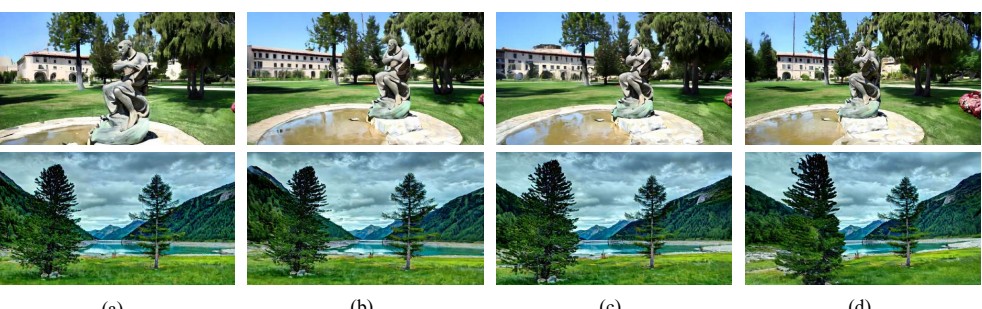

Figure L-9: Visual comparison of single view-based NVS results utilizing weight function as (a) constant, (b) Linear, (c) Exponential, (d) Ours (Post).

## M    COMPARISON WITH ZERONVS

We present a comparison of 360-degree NVS comparison results in Figure M-10. The results highlight significant differences between the approaches. ZeroNVS generates videos by adhering to a specific and relatively constrained pattern, which limits the diversity and complexity of the generated scenes. In contrast, our method demonstrates the ability to produce videos with more intricate and realistic background geometry.

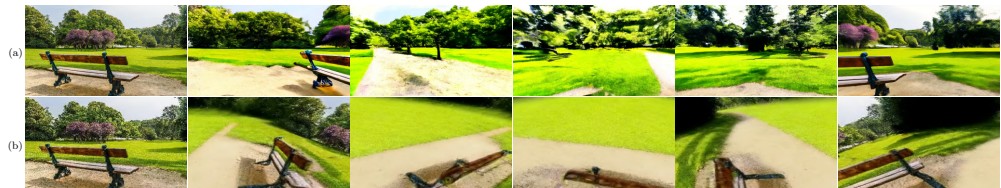

Figure M-10: Comparison on 360° NVS results between (a) Ours and (b) ZeroNVS (Sargent et al., 2024).

## N    COMPARISON WITH PHOTOCONSISTENT-NVS

We conducted a comparative experiment on Photoconsistent-NVS, and the quantitative results are presented in Figure N-11.

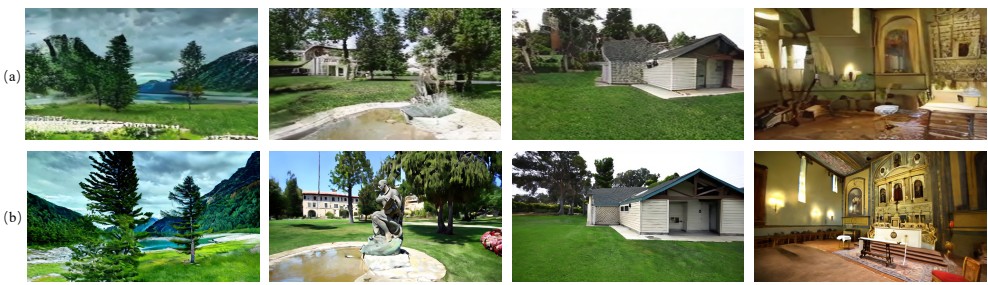

Figure N-11: Visual comparison between (a) Ours and (b) Photoconsistent-NVS (Yu et al., 2023).

## O    COMPARISON WITH SPARSE GAUSSIAN ON EXTREME ZOOM IN AND OUT

We conducted a comparative experiment by performing extreme zoom-in and zoom-out operations on Sparse Gaussian (Xiong et al., 2023), as illustrated in Figure O-12.

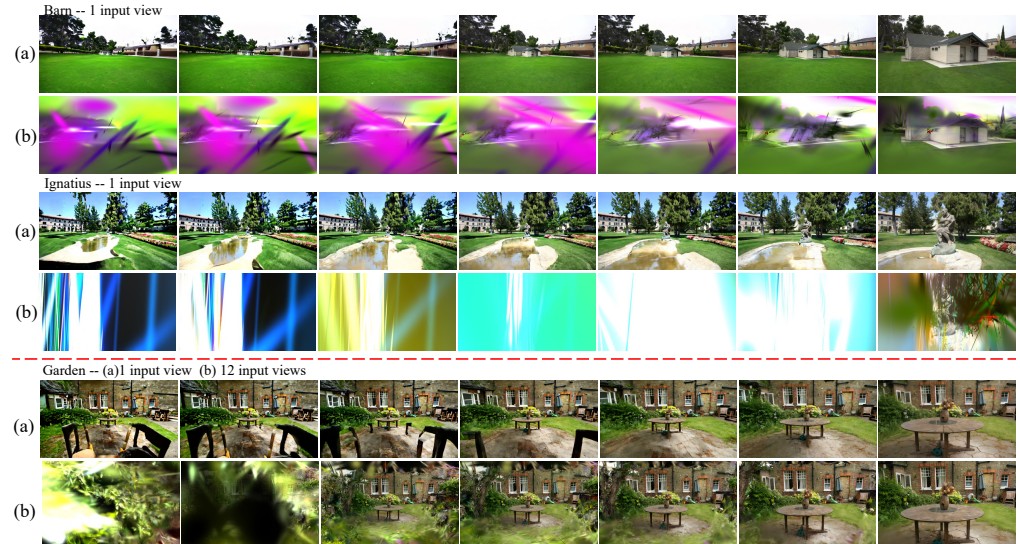

Figure O-12: Visual comparison between (a) Ours and (b) Sparse Gaussian (Xiong et al., 2023), where for the first two scenes both methods are inputted with the same one view. Moreover, we also input the 12 views to Sparse Gaussian on the third scene and keep ours with 1 scene, which is "unfair" to our method. Experimental results demonstrate the robustness and consistency of our method with large camera pose changes.

