# OpenReview forum: "NVS-Solver: Video Diffusion Model as Zero-Shot Novel View Synthesizer"
_ICLR.cc/2025/Conference — ICLR 2025 Poster_

### Official Review · Reviewer_QumE · 2024-10-27

**Soundness:** 2
**Presentation:** 2
**Contribution:** 2
**Rating:** 6
**Confidence:** 3

**Summary:**

The paper presents a training-free sampling method that generates image sequences conditioned on input camera trajectories. The input view is warped onto the target views, serving as pseudo-GT variables. To condition the reverse process, the paper re-computes the predicted mean of a reverse step with an interpolation between the predicted mean of the current noisy latent and the warped samples. It further explores calculating the optimal interpolation weight and two guidance methods (replacement and gradient). Both quantitative result and qualitative results show the proposed method can generate smooth, high-fidelity image sequences.

**Strengths:**

1. The paper presents a training-free method that enables a pre-trained diffusion model to generate image sequences based on given camera trajectories.
2. The qualitative videos demonstrate that the proposed method generates plausible outputs.

**Weaknesses:**

1. The paper relies on an off-the-shelf depth estimation network to warp the input image(s), which may be prone to scale inconsistencies across frames. Since the reverse process is largely dependent on the warped images, depth estimation errors could propagate to the final outputs.
2. The paper does not explain how occluded regions are handled in depth-warping. Handling the occluded regions can be particularly challenging when the camera pose variation is large, as $\tilde{\mathbf{\mu}}_{t, \mathbf{p}_i}$ in Eq.12 may produce degenerate outputs.
3. The camera trajectories in most qualitative results are limited to relatively small variations or 360-degree circular poses, where scale ambiguity from the depth estimation network can largely be ignored. This raises robustness of the proposed method when the camera variations are large. The paper could benefit from including more examples with larger camera trajectories.
4. The toy experiment in Fig.2 (a) shows that the $\mathcal{E}_D$ decreases with the diffusion reverse process. This brings another question on how the predicted means converge to the GT image (loss=0) where it is very unlikely to have sampled $\mathbf{X}_t$ that would have led to the desired GT. I believe more details of the experiment can help understanding the paper better.
5. The paper makes some assumptions to compute the optimal interpolation weight $\lambda(t, \mathbf{p}_i)$ in Sec.4.2. However, the paper does not present ablation study on the choice of the weight (except when $\lambda(t, \mathbf{p}_i) = \infty$, which also shows comparable results). To validate the choice of the weight schedule, the paper could present comparisons to other naive techniques (e.g., linear, constant, exponential). Additionally, the interpolation weight depends on a set of hyperparameters $\{ v_1, v_2, v_3 \}$ which may require engineering effort to tune on new scenes.
6. While the paper shows promising quantitative and qualitative results, the number of scenes used for evaluation is insufficient to validate the method's effectiveness and differs from previous state-of-the-art methods.

**Questions:**

1. The proposed method sets the number of inference step to 100, which is quite large. Do other diffusion-based baselines also use the same number of inference step?
2. The paper mentions the camera trajectory is estimated (L.360). Could you provide more details on how the pose metrics (ATE, RPE) are computed?

---

> ### Author Response · Authors · 2024-11-22
> **Response to Reviewer QumE (Part I)**
>
> > **1. Comment: The paper relies on an off-the-shelf depth estimation network to warp the input image(s), which may be prone to scale inconsistencies across frames. Since the reverse process is largely dependent on the warped images, depth estimation errors could propagate to the final outputs.**
>
> **Response**: **Depth estimation error.**  For multiview-based depth estimation, we use COLMAP to obtain depth data, ensuring consistency between the camera pose and depth. For monocular video-based novel view synthesis (NVS), we utilize Depth-Anything, a tool designed for relative depth estimation. Depth-Anything, particularly Depth-Anything V2, represents a state-of-the-art depth estimation module capable of generating consistent depth maps across frames in a monocular video. This configuration allows us to maintain depth consistency throughout the NVS process, effectively avoiding the issue of ‘scale inconsistency.’
>
>
>
> **You may have misunderstood our core idea.** Our method is **exactly** designed to solve that depth estimation error. While it is true that depth estimation introduces errors, we have explicitly addressed this issue in Sections 4.1 and 4.2. Our NVS-oriented adaptive score modulation is specifically designed to mitigate the impact of these errors. By adaptively modulating the prior of the given view within the diffusion process, our approach reduces the influence of depth inaccuracies. The warped images serve as a coarse condition, which is then further refined and adjusted by the diffusion model in an adaptive manner. Moreover, **our method is highly versatile and can effectively adapt to different modern depth estimation techniques**. As demonstrated in Table F-4 of the manuscript, our approach consistently achieves strong performance across various depth estimation methods, outperforming the state-of-the-art NVS method, MotionCtrl.
>
> Furthermore, the field of depth estimation has made significant strides in recent years, with increasingly advanced models delivering higher levels of accuracy and robustness. Modern depth estimation methods now effectively handle a wide range of scenarios, from complex outdoor landscapes to dynamic indoor scenes, with greater precision and consistency. These advancements are unlocking new possibilities for reliable and versatile applications in areas such as autonomous driving, augmented reality, and novel view synthesis. As the accuracy of these methods continues to improve, the performance of our approach is expected to advance correspondingly.
>
> **Last but not least**, it is worth noting that in the era of deep learning, the performance of large models tailored for specific tasks has significantly improved due to the surge in data and computing power. Utilizing pre-trained off-the-shelf models as one component of a multi-stage processing system to be constructed is becoming a prominent and emerging trend.  The proposed framework's inherent ability to seamlessly incorporate pre-trained models ensures its adaptability and positions it to effectively leverage insights from future, more sophisticated models.
>
>
>
> > **2. Comment: The paper does not explain how occluded regions are handled in depth-warping. Handling the occluded regions can be particularly challenging when the camera pose variation is large, as in Eq.12 may produce degenerate outputs.**
>
> **Response**: **Occlusion.** We believe you may have **misunderstood** certain aspects of our method. For occluded regions, the forward warp operation creates holes in the target image. These holes, which lack valid values, are masked and subsequently generated solely through the diffusion process. The operation described in Eq. (12) is applied exclusively to the unmasked regions containing valid values.
>
> > **3. Comment: The camera trajectories in most qualitative results are limited to relatively small variations or 360-degree circular poses, where scale ambiguity from the depth estimation network can largely be ignored. This raises the robustness of the proposed method when the camera variations are large. The paper could benefit from including more examples with larger camera trajectories.**
>
> **Response**: **Larger camera trajectories.**  We believe you may have **overlooked** our experimental section on 360-degree configurations. As illustrated in Figure 5, we present two setups: (a) the camera remains stationary at the center of the scene and rotates 360 degrees, and (b) the camera moves along a 360-degree circular trajectory while continuously facing the center of the scene. Configuration (a) aligns with what you referred to as a "360-degree circular pose," whereas configuration (b) involves an extremely large trajectory, where the camera follows a 360-degree circular path.

---

> ### Author Response · Authors · 2024-11-22
> **Response to Reviewer QumE (Part II)**
>
> > **4. Comment: The toy experiment in Fig.2 (a) shows that the decreases with the diffusion reverse process. This brings another question on how the predicted means converge to the GT image (loss=0) where it is very unlikely to have sampled that would have led to the desired GT. I believe more details of the experiment can help understanding the paper better.**
>
> **Response**: **The mean value in Fig.2 (a).** We believe you may have **misunderstood** the mean value curve in Figure 2(a). This curve represents the averaged F2 Norm of estimation errors across different noise levels but not the average value of any predictions. It only has statistical meaning; rather, it is included to illustrate the trend of how diffusion estimation errors decrease as the noise level changes.
>
> > **5. Comment: The paper makes some assumptions to compute the optimal interpolation weight in Sec.4.2. However, the paper does not present ablation study on the choice of the weight (except when, which also shows comparable results). To validate the choice of the weight schedule, the paper could present comparisons to other naive techniques (e.g., linear, constant, exponential). Additionally, the interpolation weight depends on a set of hyperparameters which may require engineering effort to tune on new scenes.**
>
> **Response**: **Ablation study on the choice of the weight and hyperparameters.** Following your suggestion, we conducted ablation experiments to evaluate different weight function choices, including (1) a constant value of 0.5
>     \begin{equation}
>         \widehat{\lambda}(t,\mathbf{p}_i) = 0.5,
>     \end{equation}
>     (2) a linear function
>     \begin{equation}
>         \widehat{\lambda}(t,\mathbf{p}_i) = t,
>     \end{equation}
>     (3) an exponential function
>     \begin{equation}
>         \widehat{\lambda}(t,\mathbf{p}_i) = e^{(t+1)}.
>     \end{equation} The quantitative and qualitative results results are presented in the table below and Fig. L-9 in Appendix. These results highlight the superiority of our weight design, as all three simple weight functions lead to decreased performance in generated image quality and trajectory accuracy, particularly when using a constant value for weighting. Additionally, as discussed in Section 5, we used the same set of values (v1, v2, v3) across all nine scenes for each setting. These hyperparameters remain consistent and do not require adjustment for different scenes.
>
>
> | Methods  | FID   | ATE  | RPE-T  | RPE-R |
> |----------|-------|------|--------|-------|
> |constant (0.5 ) |  175.68  | 6.09|1.06| 0.864  |
> | linear | 174.23 |  1.04 |0.210| 0.199 |
> | exp | 174.60 |  1.363 |0.312| 0.330 |
> |Ours |165.12 | 0.767 | 0.156 | 0.170 |
>
> > **6. Comment: While the paper shows promising quantitative and qualitative results, the number of scenes used for evaluation is insufficient to validate the method's effectiveness and differs from previous state-of-the-art methods.**
>
> **Response**: **Number of scenes.** We would like to highlight some well-known view synthesis methods here. First, the NeRF method used three datasets: the DeepVoxel synthetic dataset (4 scenes), the Blender dataset (8 scenes), and the LLFF dataset (8 scenes). Mip-NeRF 360 utilized its own 360-degree dataset, which consists of 9 scenes. 3D Gaussian Splatting was evaluated on the full set of scenes presented in Mip-NeRF 360 (9 scenes), along with two scenes from the Tanks&Temples dataset and two scenes provided by Hedman et al. As outlined in Section 5, we evaluate our method across single-view, multi-view, and dynamic-view settings, each using 9 scenes. Therefore, we believe that 9 scenes per setting are sufficient, and our choice does not result in a reduced number of scenes compared to state-of-the-art methods. We list the number of scenes as the following table.
>
> | Methods  | NeRF   | Mip-NeRF 360  | 3D GS | Ours|
> |----------|------- |------|--------|-------|
> | Num of Scene | 20     | 9    | 13      | 27|
>
>
> > **7. Comment: The proposed method sets the number of inference step to 100, which is quite large. Do other diffusion-based baselines also use the same number of inference step?**
>
> **Response**: **Number of inference step.** The inference step is a tunable hyperparameter that you can adjust. As illustrated in Section 5.3, we did ablation on reverse inference steps from 25 to 100. Using a smaller value can speed up inference. the synthesized image quality of our method does not improve intensely with the number of inference steps increasing. However, the pose error decreases significantly, indicating the necessity of a sufficient number of inference steps for accurately rendering novel views. 3D-Aware employs a sequential unconditional-conditional multiview image generation process, with the default number of sampling steps set to 1000 for the unconditional model and 50 for the conditional model. In MotionCtrl, the default number of sampling steps is 25.

---

> ### Author Response · Authors · 2024-11-22
> **Response to Reviewer QumE (Part III)**
>
> > **8. Comment: The paper mentions the camera trajectory is estimated (L.360). Could you provide more details on how the pose metrics (ATE, RPE) are computed?**
>
> **Response**: **Pose metrics.** Following MotionCtrl, we use Particle-SFM (https://github.com/bytedance/particle-sfm) to estimate and evaluate the camera trajectory for generated images. In Particle-SFM, pose error calculation is performed using the evo_ape and evo_rpe commands from the evo tool (https://github.com/MichaelGrupp/evo). Although Particle-SFM cannot recover the real-world scale of the scene from the video input, the pose error calculation process in evo first normalizes the ground truth trajectories to unit length, then aligns the predicted trajectories to the normalized ground truth using Umeyama alignment with scale calibration before calculating the metrics. We will provide more details on the pose metrics computation in the manuscript.

---

> ### Author Response · Authors · 2024-11-25
>
> Hi, Reviewer QumE. We have conducted empirical experiments and provided theoretical explanations to resolve your concerns. Feel free to discuss if you have any further concerns.

---

> > ### Comment · Reviewer_QumE · 2024-11-26
> >
> > I appreciate the authors for addressing the concerns.
> >
> > While it is not a reason to reject the paper, I still find the "depth scale" aspect unconvincing, which raises concerns about the method's applicability to free-pose camera trajectories.
> > The authors claim that recent off-the-shelf depth predictors show state-of-the-art performances, but depth scale ambiguity of single view image is an inherently ill-posed problem. The paper does not mention any of these limitations and the results focus on the examples where scale ambiguity factor can be largely ignored (small variations, object centric 360 rotation and panoramic trajectory).

---

> > > ### Author Response · Authors · 2024-11-26
> > >
> > > Hi, Reviewer QumE. To address your concerns further, we conducted additional experiments, detailed in **Fig. O-12** of the Appendix. These experiments explore scenarios beyond the mentioned cases (e.g., small variations, object-centric 360° rotation, and panoramic trajectories). The results validate that the proposed method effectively handles significant camera variations.
> > >
> > > Furthermore, given the inherent limitations of single-view images, we acknowledge that **deriving the absolute scale is a challenging and, in many cases, intractable task** due to the ill-posed nature. Even specially designed methods struggle to address this issue. This is not a limitation unique to the proposed method **but rather a common challenge** faced by all NVS (Novel View Synthesis) and depth estimation algorithms.

---

> > > ### Author Response · Authors · 2024-11-29
> > >
> > > Hi, Reviewer QumE. Moreover, we would like to clarify that **both SparseGaussian and SparseNeRF employ depth maps from pre-trained depth models** as priors in their methods. This further demonstrates that incorporating pre-trained depth models is a common and effective approach for supporting novel view synthesis.

---

> ### Author Response · Authors · 2024-11-28
>
> Hi **Reviewer QumE**,
>
> We have conducted further experiments to address your concerns. We believe the aforementioned theoretical analysis, along with the empirical studies, highlights the distinctions between our methods and other SOTA methods.

---

> ### Author Response · Authors · 2024-11-29
>
> Hi **Reviewer QumE**,
>
> We have made every effort to address your concerns and would greatly appreciate any additional feedback you may have. Given that there are only a few days remaining for the discussion, we would be truly grateful if you could take a few minutes to review our response. Thank you for your time and consideration!

---

> ### Author Response · Authors · 2024-12-02
>
> Hi **Reviewer QumE**,
>
> We have made every effort to address all the issues and comments provided in a thorough and effective manner. We kindly expect that the reviewer can take a moment to review our response and share any feedback or reconsider your rating. Your time and thoughtful consideration are greatly appreciated.
>
> Best regards,
>
> The authors.

---

> > ### Comment · Reviewer_QumE · 2024-12-02
> >
> > Thank you for the clarifications. My concerns are addressed, and I raise my score to positive.

---

> > > ### Author Response · Authors · 2024-12-02
> > >
> > > The authors sincerely appreciate your reply.

---

### Official Review · Reviewer_b97v · 2024-10-30

**Soundness:** 2
**Presentation:** 3
**Contribution:** 2
**Rating:** 6
**Confidence:** 4

**Summary:**

The authors propose a training free novel view synthesis paradigm based on video diffusion model. Specificcally, warped depth maps are utilized and certain sampling methods are proposed to ensure high quality NVS.

**Strengths:**

1. The work is complete and results seem to be good quantitively.

**Weaknesses:**

1. The work is complete, but the novelty and the contribution are not strong enough to meet the quality of ICLR. A good training-free is not that appealing after previous works like MotionCtrl, and it highly relys on the capability of SVD. It is hard to know about it generalizability around various video generation methods.
2. It is recommended to show "Directly guided sampling" and "Posterior sampling" clearly in the view of practical implementation. Some illustration would be appreciated.
3. Some equations should be displayed more clearly. For instance in eq.8, I(P_0) should not be related to the pose index i, but in eq.9 it is related to i. Maybe in eq.9 the I(P_0) should be revised as I(P_i)?

**Questions:**

Please refer to Weaknesses.

---

> ### Author Response · Authors · 2024-11-22
> **Response to Reviewer b97v**
>
> > **1. Comment:  The work is complete, but the novelty and the contribution are not strong enough to meet the quality of ICLR. A good training-free is not that appealing after previous works like MotionCtrl, and it highly relys on the capability of SVD. It is hard to know about it generalizability around various video generation methods.**
>
> **Response**: **Generalization ability around various video generation methods.** Our method introduces an adaptive strategy at each sampling step, offering a novel enhancement that can, in theory, be **integrated with any diffusion-based approach**. This flexibility underscores the potential generalization ability of our technique across a wide range of diffusion models. We utilize SVD, due to that it's the most stable and effective choice.
>
> Moreover, our current implementation **already significantly outperforms state-of-the-art methods** in both image generation quality and pose accuracy metrics. This demonstrates the robustness of our approach and its capacity to deliver superior outcomes, underscoring the effectiveness and necessity of building the proposed adaptive strategy in advancing diffusion-based modeling.
>
> In future work, we plan to extend and validate our method within other diffusion frameworks, further demonstrating its versatility and applicability.
>
>
> > **2. Comment: It is recommended to show "Directly guided sampling" and "Posterior sampling" clearly in the view of practical implementation. Some illustration would be appreciated.**
>
> **Response**: **Illustration between "Directly guided sampling" and "Posterior sampling".** Eq. (13) and Eq. (14) indeed illustrate how we implement "Directly Guided Sampling (DGS)" and "Posterior Sampling (POST)". In the code, DGS is achieved by directly **updating/replacing** the predicted values in the masked regions to align with the latent features. The POST process **computes the loss** based on the difference between the predicted sample and the conditional latent features, with the loss weighted by the mask. It then **back-propagate** gradients of this loss to the sample to optimize sample adapt to the given scene.
>
> While DGS provides a straightforward approach for solving reverse stochastic differential equations, it is limited to standard, unconditional diffusion processes. In contrast, POST introduces a more sophisticated mechanism for conditional generation, utilizing dynamic masking and feature-specific refinement to align the generated samples with external latent guidance. This added complexity, although computationally more intensive, makes Post highly suitable for tasks requiring precise feature alignment or controlled generation, thus showing its potential for more specialized generation in novel view synthesis.
>
>
> > **3. Comment: Some equations should be displayed more clearly. For instance in eq.8, I(P_0) should not be related to the pose index i, but in eq.9 it is related to i. Maybe in eq.9 the I(P_0) should be revised as I(P_i)?**
>
> **Response**: **Equations typo.** We believe you may have **misunderstood**  Eq. (8) and Eq. (9). We actually applied the Taylor expansion of $\mathcal{I}( \mathbf{p}_i)$ on the point of $\mathbf{p}_0$. And the $\mathbf{p}_0$ in the Eq. (9) indicates to calculate the **zero-order term** $\mathcal{I}( \mathbf{p}_0)$ in Eq.(8). Specifically, Eq. (8) formulates the calculation of $\mathcal{I}( \mathbf{p}_i)$ via building upon the known intensity at the reference pose $ \mathbf{p}_0 $ and leveraging the geometric transformation induced by the relative pose variation $\Delta \mathbf{p} = \mathbf{p}_i - \mathbf{p}_0$ and the depth information $\hat{\mathbf{D}}$. Eq. (9) projects the reference image to the target pose through the warping operation $\mathcal{W}(\cdot)$.

---

> ### Author Response · Authors · 2024-11-25
>
> Hi, Reviewer b97v. We have provided theoretical explanations to resolve your concerns. Feel free to discuss if you have any further concerns.

---

> ### Author Response · Authors · 2024-11-28
>
> Hi **Reviewer b97v**,
>
> In the previous reply, we have clearly clarified your concerns. Moreover, we believe the theoretical analysis, along with the empirical studies in our paper, highlights the distinctions between our methods and other methods.

---

> > ### Author Response · Authors · 2024-11-29
> >
> > Hi **Reviewer b97v**,
> >
> > We have made every effort to address your concerns and would greatly appreciate any additional feedback you may have. Given that there are only a few days remaining for the discussion, we would be truly grateful if you could take a few minutes to review our response. Thank you for your time and consideration!

---

> ### Author Response · Authors · 2024-12-02
>
> Hi **Reviewer b97v**,
>
> We have made every effort to address all the issues and comments provided in a thorough and effective manner. We kindly expect that the reviewer can take a moment to review our response and share any feedback or reconsider your rating. Your time and thoughtful consideration are greatly appreciated.
>
> Best regards,
>
> The authors.

---

### Official Review · Reviewer_T9eW · 2024-11-03

**Soundness:** 3
**Presentation:** 2
**Contribution:** 3
**Rating:** 6
**Confidence:** 3

**Summary:**

1. This paper proposes a novel view synthesis pipeline without any training. The pipeline can take single or multiple views of static scenes or monocular videos of dynamic scenes as input.
2. This paper modulates the score function with the warped input views to control the video diffusion process and generate visually pleasing results. They achieve the modulation in an adaptive fashion based on the view pose and the number of diffusion steps.
3. They conduct extensive results on both static and dynamic scenes and show promising results with both evaluation numbers and visualizations.

**Strengths:**

1. The proposed adaptive modulation of the score function in the diffusion process is novel.
2. The proposed method achieves better results in various scenarios compared to baselines.
3. The authors provide the code with an anonymous link, ensuring the applicability of the results.

**Weaknesses:**

My primary concerns are with the references and experimental details:

1. Some key references on diffusion-based NVS are missing [1,2,3,4,5,6,7]. Among these, [3] specifically focuses on scenes and has released its code. Is there a particular reason it was not included in the comparison?
2. How is the synthesized view pose calculated in this paper? In Line 364, it states that 'current depth estimation algorithms struggle to derive absolute depth from a single view or monocular video, resulting in a scale gap between the synthesized and ground truth images.' When calculating pose error, does the proposed method account for this scale gap?

**Minor Points:**

1. In Table 1, it is stated that MotionCtrl [Wang et al., 2023b] and 3D-aware [Xiang et al., 2023] do not require training. However, as I understand, they do require fine-tuning.

[1] Wu, Rundi, et al. "Reconfusion: 3d reconstruction with diffusion priors." *Proceedings of the IEEE/CVF Conference on Computer Vision and Pattern Recognition*. 2024.

[2] Watson, Daniel, et al. "Novel View Synthesis with Diffusion Models." *The Eleventh International Conference on Learning Representations*.

[3] Yu, Jason J., et al. "Long-term photometric consistent novel view synthesis with diffusion models." *Proceedings of the IEEE/CVF International Conference on Computer Vision*. 2023.

[4] Cai, Shengqu, et al. "Diffdreamer: Towards consistent unsupervised single-view scene extrapolation with conditional diffusion models." Proceedings of the IEEE/CVF International Conference on Computer Vision. 2023.

[5] Tseng, Hung-Yu, et al. "Consistent view synthesis with pose-guided diffusion models." *Proceedings of the IEEE/CVF Conference on Computer Vision and Pattern Recognition*. 2023.

[6] Chan, Eric R., et al. "Generative novel view synthesis with 3d-aware diffusion models." *Proceedings of the IEEE/CVF International Conference on Computer Vision*. 2023.

**Questions:**

Line 415 mentions that the proposed method can achieve 360-degree NVS. Would it be possible to include a comparison with ZeroNVS [7] to better demonstrate its effectiveness?


[7] Sargent, Kyle, et al. "ZeroNVS: Zero-Shot 360-Degree View Synthesis from a Single Image." *Proceedings of the IEEE/CVF Conference on Computer Vision and Pattern Recognition*. 2024.

---

> ### Author Response · Authors · 2024-11-22
> **Response to Reviewer T9eW**
>
> > **1. Comment: Some key references on diffusion-based NVS are missing [1,2,3,4,5,6,7]. Among these, [3] specifically focuses on scenes and has released its code. Is there a particular reason it was not included in the comparison?**
>
> **Response**: Thanks for your suggestion. We'll add all these diffusion-based NVS methods to our related work.
> We conducted a comparative experiment on Photoconsistent-NVS, and the qualitative results are presented in the table below. We show the quantitative results in Figure N-11. Our method outperforms Photoconsistent-NVS, particularly in terms of the accuracy of the generated trajectory.
> For the synthesize trajectory setup to generate images, we generate a series of camera positions and orientations that move around a central point, ensuring each camera consistently focuses on the scene's center. This approach allows the cameras to traverse a smoothly upward-curving path while maintaining an upright orientation. Depending on the setup, the cameras can move in either a clockwise or counterclockwise direction around the scene. By doing so, we capture the environment from multiple perspectives, providing comprehensive spatial coverage and seamless transitions between different viewpoints. This ensures a dynamic and cohesive exploration of the scene, enhancing the overall visualization quality.
>
> > **2. Comment: How is the synthesized view pose calculated in this paper? In Line 364, it states that 'current depth estimation algorithms struggle to derive absolute depth from a single view or monocular video, resulting in a scale gap between the synthesized and ground truth images.' When calculating pose error, does the proposed method account for this scale gap?**
>
> **Response**: Following MotionCtrl, we use Particle-SFM (https://github.com/bytedance/particle-sfm) to estimate and evaluate the camera trajectory for generated images. In Particle-SFM, pose error calculation is performed using the evo_ape and evo_rpe commands from the evo tool (https://github.com/MichaelGrupp/evo). Although Particle-SFM cannot recover real-world scale of the scene from the video input, the pose error calculation process in evo first normalizes the ground truth trajectories to unit length, then aligns the predicted trajectories to the normalized ground truth using Umeyama alignment with scale calibration before calculating the metrics. We will provide more details on the pose metrics computation in the manuscript.
>
> | Methods  | FID   | ATE  | RPE-T  | RPE-R |
> |----------|-------|------|--------|-------|
> |Ours |165.12 | 0.767 | 0.156 | 0.170 |
> |Photoconsistent-NVS | 193.87 | 7.64 | 1.19 | 1.45|
>
> > **3. Comment:  In Table 1, it is stated that MotionCtrl [Wang et al., 2023b] and 3D-aware [Xiang et al., 2023] do not require training. However, as I understand, they do require fine-tuning.**
>
> **Response**: **Necessity of fine-tuning for MotionCtrl/ 3D-aware.** Sorry for the misunderstanding in the Table 1. Here, "train" does not refer to the absence of fine-tuning; rather, we mean whether the method requires training when given a new sample (scene-specific training). A more accurate term would be "overfitting." NeRF- and Gaussian-based methods share a common characteristic: they require training on the specific images of a scene to fit the data. In contrast, diffusion-based methods do not require any additional training when performing inference on a given scene.
>
> > **4. Comment:  Line 415 mentions that the proposed method can achieve 360-degree NVS. Would it be possible to include a comparison with ZeroNVS [7] to better demonstrate its effectiveness?**
>
> **Response**: **Comparison with ZeroNVS.** We present a comparison of 360-degree NVS comparison results in **Figure M-10**. The results highlight significant differences between the approaches. ZeroNVS generates videos by adhering to a specific and relatively constrained pattern, which limits the diversity and complexity of the generated scenes. In contrast, our method demonstrates the ability to produce videos with more intricate and realistic background geometry.

---

> > ### Comment · Reviewer_T9eW · 2024-11-23
> >
> > I appreciate the author's response. They added the missing references and conducted experiments to compare them with Photoconsistent-NVS. The response has addressed my concerns.

---

> > > ### Author Response · Authors · 2024-11-23
> > > **Thanks for the Reply.**
> > >
> > > The authors sincerely appreciate your reply.

---

### Official Review · Reviewer_cA11 · 2024-11-03

**Soundness:** 3
**Presentation:** 3
**Contribution:** 3
**Rating:** 6
**Confidence:** 4

**Summary:**

This paper introduces a NVS method that leverages large pre-trained video diffusion models without additional training. The approach adaptively modulates the diffusion sampling process using input views to produce high-quality results from single or multiple views of static scenes or dynamic videos. Theoretical modeling is used to iteratively adjust the score function based on scene priors, enhancing control over the diffusion process. The modulation adapts to view pose and diffusion steps.

**Strengths:**

1. The proposed approach is entirely training-free, meaning it directly leverages pre-trained large video diffusion models without requiring additional fine-tuning or retraining. This feature not only reduces computational demands but also makes it adaptable to a wide range of applications where time or resources for training may be limited. The flexibility of using pre-trained models enhances its practicality, allowing users to apply this method to various scenes and tasks with minimal setup.

2. The generated videos maintain high visual fidelity, delivering smooth results. This quality stems from the adaptive modulation of the diffusion process, which effectively incorporates scene details and structures from the given views, ensuring that outputs are kind of realistic.

3. The method is underpinned by theoretical modeling, which guides its adaptive modulation strategy. By iteratively adjusting the score function with scene priors and analyzing estimation error boundaries, the approach achieves both controlled and adaptive modulation of the diffusion process.

**Weaknesses:**

1. The comparison between this method and NeRF-based methods is fundamentally imbalanced. NeRF techniques incorporate an underlying 3D structure, enabling them to render any view with predictable performance, as the 3D structure informs which views are feasible and which are not. In contrast, the proposed method lacks an explicit 3D representation, limiting its view synthesis capabilities to specific views with no guarantee of consistent performance. This distinction is significant, as NeRF's inherent 3D information allows interpretable, reliable results across views, whereas this method’s output reliability is less predictable and may vary based on input views.

2. To achieve a fair comparison between the proposed method and NeRF-based techniques, the authors should first reconstruct a 3D model from the output video of this method and then re-render the scene from that reconstructed model. This process would allow for a direct assessment of both methods’ rendering consistency and quality, ensuring that comparisons consider the 3D structure NeRF inherently leverages. Also, reconstruction error like PSNR should be reported.

3. The video results of the proposed method exhibit visible flickering artifacts, which could substantially affect reconstruction quality and consistency. A deeper analysis is needed to assess how these artifacts impact overall reconstruction accuracy and to identify potential mitigation strategies. This might include tuning reconstruction parameters to minimize flickering, which would help improve the method’s output stability and robustness, especially for applications sensitive to temporal consistency.

4. A major contribution is the derivation of the parameter $\lambda$ in Section 4.2, which aims to minimize the estimation error upper bound in Equation 15. However, a gap remains between this upper bound and the actual estimation error represented in the left side of Equation 15. To strengthen the theoretical foundation, the authors should provide a more comprehensive analysis of how reducing the upper bound affects the actual estimation error. This could be achieved through statistical analysis and empirical evidence showing how well the method reduces estimation error in practice, thereby validating the theoretical assumptions.

**Questions:**

For dynamic scene comparison, it is said "For monocular video-based NVS, we downloaded nine videos from YouTube, each comprising
frames and capturing complex scenes in both urban and natural settings."

Why not just following the dynamic nerf settings? They have well aligned ground-truth for measuring the reconstruction performance.
Generation metrics like FID are not that reliable.

Soma example datasets are HyperNerf, DyCheck (https://hangg7.com/dycheck/)

---

> ### Author Response · Authors · 2024-11-22
> **Response to Reviewer cA11 (Part I)**
>
> > **1. Comment: The comparison between this method and NeRF-based methods is fundamentally imbalanced. NeRF techniques incorporate an underlying 3D structure, enabling them to render any view with predictable performance, as the 3D structure informs which views are feasible and which are not. In contrast, the proposed method lacks an explicit 3D representation, limiting its view synthesis capabilities to specific views with no guarantee of consistent performance. This distinction is significant, as NeRF's inherent 3D information allows interpretable, reliable results across views, whereas this method’s output reliability is less predictable and may vary based on input views.**
>
> **Response**: We agree that our method has significant distinctions compared to NeRF-based methods. However, we respectfully **disagree** with your judgment that the comparison between our method and NeRF-based methods is fundamentally imbalanced. The reviewer pointed out that NeRF-based methods, once optimized, can produce explicit 3D geometry and render any view with photometric consistency.
>
> In contrast, while our method does not reconstruct explicit geometry, the warping process ensures photometric consistency for the visible content in input images when generating target views.     Additionally, by leveraging the pre-trained video diffusion model, our approach can generate meaningful content in occluded areas and perform view extrapolation. Moreover, due to the manifold preserving sampling characteristics, **the generated video lies in the same inherent manifold with the training data**. Thus, the view consistency of the generated video is theocratically coherent with training video, which is naturally view-consistent.
>
> Furthermore, our adaptive diffusion modulating strategy ensures that our method is not inherently less predictable. While variability may arise based on input views, it is limited to areas that are occluded or outside the input view—regions that cannot be generated by NeRF-based methods. Thus, our approach offers unique advantages in handling occlusions and unseen regions, presenting a complementary perspective rather than being inherently inferior or imbalanced compared to NeRF-based methods.
>
> > **2. Comment: To achieve a fair comparison between the proposed method and NeRF-based techniques, the authors should first reconstruct a 3D model from the output video of this method and then re-render the scene from that reconstructed model. This process would allow for a direct assessment of both methods’ rendering consistency and quality, ensuring that comparisons consider the 3D structure NeRF inherently leverages. Also, reconstruction error like PSNR should be reported.**
>
> **Response**: **Qualitatively comparison with NeRF-based methods.** We do not believe that the comparison between our proposed method and NeRF-based techniques is unfair. For both methods, we use the same input data and estimate novel views based on identical trajectories. Unlike NeRF-based techniques, which require training on a specific scene, our method is training-free, allowing us to generate novel views directly without the need for any scene-specific training. Additionally, our approach offers the flexibility to generate videos from arbitrary trajectories, eliminating the need to retrain a NeRF model for each new view or re-rendering task. This makes our method both more efficient and adaptable compared to traditional NeRF-based techniques.
>
>
> We would like to clarify the misunderstanding regarding why we did not report reconstruction errors such as PSNR. The reason is the presence of a scale gap between our synthesized images and the ground truth, as explained in Section 5. This issue arises because current depth estimation algorithms struggle to derive absolute depth from a single view or monocular video. Even if we reconstruct a 3D model from the output video of our method, the scale problem persists, and the resulting 3D geometry may not align with the absolute scale, making pixel-wise reconstruction errors impossible to compute.
>
> Furthermore, while Gaussian- and NeRF-based comparison methods are generalizable approaches with sparse inputs, they are trained using ground truth images. This training allows rescale information to be incorporated at the feature level. In contrast, our method focuses on being "training-free" and does not rely on ground truth data for model training. Additionally, we have included comparisons using paired metrics such as LPIPS, as detailed in Appendix I.

---

> > ### Comment · Reviewer_cA11 · 2024-11-25
> > **still think the comparison between NeRF-based is imbalanced**
> >
> > I am not convinced by the explanation.
> >
> > NeRF-based methods are well tolerant to camera positions. And can always render perfect videos.
> >
> > It is not about the metrics on some selected "in-domain" novel views. The metrics are biased.
> >
> > If we adjust the testing set to be with a very large distance variation, like from very far to close, NeRF-based method will always be good.
> >
> > You claimed that "the view consistency of the generated video is theocratically coherent with training video" -- any support that this will be true if we test with some extreme cases? Like from very far, the object will just be a single pixel on the screen. What will the diffusion model give if there is only one wrapping pixel given? What will the diffusion give when the available wrapping pixel growing from one to 100? Will it still be multiview consistent?

---

> ### Author Response · Authors · 2024-11-22
> **Response to Reviewer cA11 (Part II)**
>
> > **3. Comment:  The video results of the proposed method exhibit visible flickering artifacts, which could substantially affect reconstruction quality and consistency. A deeper analysis is needed to assess how these artifacts impact overall reconstruction accuracy and to identify potential mitigation strategies. This might include tuning reconstruction parameters to minimize flickering, which would help improve the method’s output stability and robustness, especially for applications sensitive to temporal consistency.**
>
> **Response**: **Flickering artifacts.** Firstly, as we adopt SVD model for the diffusion process, flickering artifact is potentially stemmed from the limited capability of the SVD. We will try more effective diffusion models to validate their performance. With the development of large video generation models, their ability to produce smoother and more realistic results continues to improve.
>
> Furthermore, since we obtain our adaptive $\gamma(t, p_i)$ by minimizing the expected value of the estimation error, this can lead to fluctuations in performance on specific data. Therefore, to achieve better results, hyperparameters can be adapted for the specific scenes.
>
> Finally, from the perspective of image/video process, such flickering artifacts can be further reduced by such prior terms link spatial temporal smoothness, or low-rankness/ sparsity of inherent data structure.
>
>
> > **4. Comment:  A major contribution is the derivation of the parameter in Section 4.2, which aims to minimize the estimation error upper bound in Equation 15. However, a gap remains between this upper bound and the actual estimation error represented in the left side of Equation 15. To strengthen the theoretical foundation, the authors should provide a more comprehensive analysis of how reducing the upper bound affects the actual estimation error. This could be achieved through statistical analysis and empirical evidence showing how well the method reduces estimation error in practice, thereby validating the theoretical assumptions.**
>
> **Response**: **Effectivity of minimizing the estimation error upper bound.** To further resolve your concern, we have empirically validate the effectiveness of the proposed strategy for minimizing error upper bound. Since current the modulation weights of $\hat{\lambda}$ is derived by optimizing the upper-bound, we conduct further experiments with constant, linear, and exponential manner of parametrization of $\hat{\lambda}$, whose error bound is higher than our parametrization. Experimental results shown in the following table indicates that our methods with lower upper-bound greatly outperform other methods. This clearly suggests that our proposed weighting strategy with theocratically optimal score error upper bound is essential for achieving the optimal performance.
>
> | Methods  | FID   | ATE  | RPE-T  | RPE-R |
> |----------|-------|------|--------|-------|
> |Ours |165.12 | 0.767 | 0.156 | 0.170 |
> |constant (0.5 ) |  175.68  | 6.09|1.06| 0.864  |
> | linear | 174.23 |  1.04 |0.210| 0.199 |
> | exp | 174.60 |  1.363 |0.312| 0.330 |

---

> ### Author Response · Authors · 2024-11-22
> **Response to Reviewer cA11 (Part III)**
>
> > **5. Comment: For dynamic scene comparison, it is said "For monocular video-based NVS, we downloaded nine videos from YouTube, each comprising frames and capturing complex scenes in both urban and natural settings." Why not just following the dynamic nerf settings? They have well aligned ground-truth for measuring the reconstruction performance. Generation metrics like FID are not that reliable. Soma example datasets are HyperNerf, DyCheck (https://hangg7.com/dycheck/)**
>
> **Response**: **DyCheck dataset.** We would like to emphasize that the goal of our "Zero-shot Novel View Synthesis" method is to synthesize multiple views of a scene from an arbitrary input image or video, whether it depicts outdoor landscapes or moving animals. YouTube videos, with their greater diversity and complexity, present an even more challenging scenario. The experimental results presented in the manuscript validate that our method is both efficient and generalizable, achieving superior performance. Additionally, our method requires only an image or monocular video as input, without the need for precise camera pose information—unlike most NeRF-based methods. As discussed in Section 5, current depth estimation algorithms struggle to derive absolute depth from a single view or monocular video, leading to a scale gap between the synthesized and ground truth images. As a result, we cannot compute conventional image metrics like PSNR or SSIM against ground truth images. Moreover, the two datasets mentioned, HyperNeRF and DyCheck, primarily capture scenes focused on specific objects, such as peeling a banana in the HyperNeRF dataset and waving a teddy bear in the DyCheck dataset.
>
> Additionally, we conducted comparison experiments on three scenes from the Dycheck dataset following your suggestion. The table below illustrates the results of our method alongside two Gaussian-based methods. The three scenes from Dycheck were captured using two cameras, each positioned at a considerable distance from the other. In our experiments, we used a monocular video from one camera as input and generated a video following the trajectory of the other camera. This setup presents a significant challenge due to the substantial pose shifts, as the input and target videos capture vastly different perspectives of the scene, making the task particularly complex. To address potential scale inconsistencies, we utilized the depth maps provided in the dataset. Our method outperformed the 4D-Gaussian approach across all three metrics.The Deformable-Gaussian method failed to produce viable results in this challenging scenario.
>
> | **dycheck**  | PSNR   | SSIM  | LPIPS  |
> |----------|-------|------|--------|
> | Ours | 15.84| 0.385 | 0.410 |
> | 4D-Gaussian | 12.68 | 0.3464 | 0.737 |
> | Deformable-Gaussian | --- failed ---|

---

> ### Author Response · Authors · 2024-11-25
>
> Hi, Reviewer cA11. We have conducted empirical experiments and provided theoretical explanations to resolve your concerns. Feel free to discuss if you have any further concerns.

---

> ### Author Response · Authors · 2024-11-26
> **Further comparisons on extreme condition**
>
> Thank you very much for your thoughtful response. Regarding the comparison metrics across different views, we would like to **respectfully emphasize** that we have made our best to optimize the performance of Nerf-based methods and fairly compare with them. However, it is hard for them to effectively handle the monocular video, sparse views, or even a single view, as discussed in the manuscript.
>
> In response to your concerns about large viewpoint variations, we have conducted additional experiments to compare the NVS results, as you suggested. Specifically, as shown in **Fig. O-12** of the Appendix, the proposed method benefits from the strong generative capability of the diffusion model and our manifold-preserving sampling approach, enabling it to handle such cases effectively. In contrast, traditional volume rendering-based methods struggle to produce visually appealing results when the test view significantly differs from the training view.

---

### Official Review · Reviewer_PHjc · 2024-11-04

**Soundness:** 3
**Presentation:** 2
**Contribution:** 3
**Rating:** 6
**Confidence:** 2

**Summary:**

This work introduces a diffusion model-based approach to achieve novel view synthesis. In particular, it leverages the depth-warped views as guidance to achieve adaptative modulation. Experiments on single-view images, multi-view images, and monocular video input-based novel view synthesis showcase the efficacy of the introduced methods.

**Strengths:**

* The idea of using depth-warped images as guidance for novel view synthesis is reasonable.
* It is interesting to see that the temporal consistent video diffusion model can be effectively reformulated to achieve geometrical consistent NVS in a training-free manner.
* Experiments on several challenging settings, including 360-degree NVS from a single view, verify the significance of the introduced method.

**Weaknesses:**

* Accessing the geometry accuracy. For the 360-degree case, e.g., the truck, it would be better to apply mesh reconstruction on the rendered views, similar to Fig. 5(b) in latentSplat [Wewer et al. ECCV 2024]. The reconstructed mesh will provide a clearer understanding of how well the rendered views maintain correct geometry.

* Pixel-aligned metrics. For the NVS task, it would be better to report comparisons with state-of-the-part methods regarding pixel-aligned metrics, e.g., PSNR and SSIM.

* Discussion with feed-forward 3DGS models. It might be interesting to see comparisons with detailed analysis between the introduced methods and those feed-forward 3DGS models, e.g., pixelSplat [Charatan et al., CVPR 2024], MVSplat [Chen et al., ECCV 2024]. And it would be better to consider adding these methods to the related work for better coverage of recent NVS works.

**Questions:**

Kindly refer to [Weaknesses].

---

> ### Author Response · Authors · 2024-11-22
> **Response to Reviewer PHjc**
>
> > **1. Comment: Accessing the geometry accuracy. For the 360-degree case, e.g., the truck, it would be better to apply mesh reconstruction on the rendered views, similar to Fig. 5(b) in latentSplat [Wewer et al. ECCV 2024]. The reconstructed mesh will provide a clearer understanding of how well the rendered views maintain correct geometry.**
>
> **Response**: **Applying mesh reconstruction.** Thank you for your suggestion. We have applied 2D Gaussian Splatting to reconstruct the mesh on the generated 360° scene. As demonstrated in the results in Figure J-8, even though our method does not rely on explicit 3D geometry as an intermediate representation, it successfully maintains geometric consistency across the generated images. This highlights the strength of our approach in preserving structural coherence while generating high-quality images, further validating the robustness and effectiveness of our proposed methodology.
>
>
> > **2. Comment: Pixel-aligned metrics. For the NVS task, it would be better to report comparisons with state-of-the-part methods regarding pixel-aligned metrics, e.g., PSNR and SSIM.**
>
> **Response**: **PSNR/SSIM metrics.** We illustrated this issue in Section 5: since current depth estimation algorithms struggle to derive absolute depth from a single view or monocular video, resulting in a scale gap between the synthesized and ground truth images, we only compare the paired metrics, regarding the visual quality, in Appendix I, such as LPIPS.
>
> Additionally, we conducted comparison experiments on three scenes from the Dycheck dataset. The table below illustrates the results of our method alongside two Gaussian-based methods. The three scenes from Dycheck were captured using two cameras, each positioned at a considerable distance from the other. In our experiments, we used a monocular video from one camera as input and generated a video following the trajectory of the other camera. This setup introduces a significant challenge due to the substantial pose shifts, as the input and target videos capture substantially different perspectives of the scene, making the task particularly difficult. To address potential scale inconsistencies, we utilized the depth maps provided in the dataset. Our method outperformed the 4D-Gaussian approach across all three metrics.The Deformable-Gaussian method failed to produce viable results in this challenging scenario.
>
> | **dycheck**  | PSNR   | SSIM  | LPIPS  |
> |----------|-------|------|--------|
> | Ours | 15.84| 0.385 | 0.410 |
> | 4D-Gaussian | 12.68 | 0.3464 | 0.737 |
> | Deformable-Gaussian | --- failed ---|
>
> > **3. Comment: Discussion with feed-forward 3DGS models. It might be interesting to see comparisons with detailed analysis between the introduced methods and those feed-forward 3DGS models, e.g., pixelSplat [Charatan et al., CVPR 2024], MVSplat [Chen et al., ECCV 2024]. And it would be better to consider adding these methods to the related work for better coverage of recent NVS works.**
>
> **Response**: **Discussion with 3DGS models.** Thank you for your suggestion. We have included a brief introduction to those methods in the related work section due to the page limit. Moreover, we will add more feed-forward 3DGS models and corresponding discussion to the related work section in the final version. PixelSplat reconstructs 3D radiance fields parameterized by 3D Gaussian primitives from pairs of images by predicting a dense probability distribution over 3D space and sampling Gaussian means from that distribution. MVSplat predicts clean feed-forward 3D Gaussians from multi-view images by incorporating a cost-volume representation, which provides valuable geometric cues to learn Gaussians. Below, we outline the key differences between these methods and ours:
>
> (1) The listed 3DGS models learn an explicit intermediate 3D model, whereas our method generates novel views based on **local Taylor expansion** of illumination function. We minimize the potential error upper bound of diffused novel views.
>
> (2) The mentioned methods rely on the inherent continuity of the kernel/ neural network to achieve novel view interpolation from input views and do not have extra knowledge than the given views. In contrast, we leverage both given views and generative models (SVD) to achieve NVS, which is naturally capable of synthesizing realistic and reasonable objects that do not exist in the given views. This is demonstrated by our superior performance in single-view and monocular video settings.
>
> (3) Although the mentioned methods are generalizable for novel view synthesis, they still require training and need plenty of given views. Our method, benefiting from the SVD model, is training-free and performs well across various types of data.
>
> (4) The mentioned methods are limited to static scene reconstruction, while our method can handle dynamic view synthesis without requiring additional techniques.

---

> ### Author Response · Authors · 2024-11-25
>
> Hi, Reviewer PHjc. We have conducted empirical experiments and provided theoretical explanations to resolve your concerns. Feel free to discuss if you have any further concerns.

---

### Author Response · Authors · 2024-11-23
**General Response**

We thank all reviewers for your time and constructive comments. Moreover, we believe all concerns have been clearly and directly addressed. Here, we also want to summarize the key clarifications concerning the contributions of our work.

(1) **Outline of the framework.** The proposed method is designed fundamentally based on **Taylor expansion of the intensity function**, then modulates the diffusion process via minimizing the potential error bound. Those errors stem from inaccuracy score estimation and high-order Taylor series residual terms ( representing the physical meaning of non-Lambertian, occlusion, and geometric/depth estimation errors).


(2) **Experiments.** We have conducted extensive experiments to validate the proposed methods across 27 scenes, from static to dynamic, and with single or multiple given views. We have added further experiments on the datasets of Dycheck with 3 scenes to further validate the proposed method. We also include an additional comparison method, Photoconsistent-NVS, for the single-view setting and provide a visual comparison with ZeroNVS in the 360-degree setting. Additionally, we reconstruct a mesh for the generated 360-degree scene to evaluate its quality. Furthermore, we conduct an ablation study on the weighting function to further demonstrate the effectiveness of our proposed strategy. **We sincerely invite you to refer to J-N of Appendix for more details.**

(3) **Novelty claim.** The proposed method is a training-free diffusion sampling algorithm, marking a fundamental divergence from existing diffusion-based NVS approaches. Furthermore, our experimental results have demonstrated the effectiveness of the proposed method in comparison to SOTA methods, e.g., MotionCtrl.

Finally, we sincerely thank the PCs, ACs, and reviewers again for your time and effort in our submission. We appreciate any further questions and discussions.

---

### Meta-Review · Area_Chair_RRTu · 2024-12-22

**Metareview:**

Summary:
- The paper presents a training-free method for novel view synthesis using a pre-trained video diffusion model. The core idea is to adaptively modulate the diffusion sampling process with the given views to ensure consistency with respect to the given views. The method is evaluated on multiple datasets and shows improved results over several baselines.

Strength:
- Interesting *training-free* approach for adapting video diffusion models for the task of novel view synthesis.
- The method can take single image, multiple images, or videos as input. The 360-degree NVS from a single view is quite interesting.

Weakness:
- Missing discussions with 3DGS methods
- Missing results on view synthesis benchmarks like DyCheck or hyperNeRF dataset.
- Temporal flickering artifacts in the video results (from the supp material).

Decision justification:
- The paper initially received borderline reviews. In the rebuttal, the authors provided a more comprehensive set of experiments (e.g., mesh reconstruction, dyCheck dataset, additional comparisons with ZeroNVS, and Photoconsistent). These additional experiments help alleviate the initial concerns. The final ratings from the reviewers are consistently positive (tho mildly only). After reading the reviews/rebuttal and checking the paper and demo video, the AC believes that while the results still have temporal flickering artifacts, its exploration of the training-free video diffusion model is interesting and could inspire future exploration in this space. The experimental validation looks solid. Therefore, the AC recommends to accept.

**Additional Comments On Reviewer Discussion:**

The primary discussions/changes of the paper were the additional evaluation:

This includes
- comparisons with ZeroNVS, and Photoconsistent-NVS evaluation in 360-degree setting.
- additional experiments on the DyCheck datset.
- ablation study on the weighting function.
- testing extreme zoom-out and zoom-in scenarios.

Overall, the reviewers appreciate the authors' detailed responses. These additional experiments adequately address their initial concerns.

---

### Decision · Program_Chairs · 2025-01-22

Accept (Poster)